# Real-Time Cost Optimization Approach Based on Deep Reinforcement Learning in Software-Defined Security Middle Platform

Yuancheng Li *[ID] and Yongtai Qin

School of Control and Computer Engineering, North China Electric Power University, Beijing 102206, China
* Correspondence: ycli@ncepu.edu.cn

**Abstract:** In today's business environment, reducing costs is crucial due to the variety of Internet of Things (IoT) devices and security infrastructure. However, applying security measures to complex business scenarios can lead to performance degradation, making it a challenging task. To overcome this problem, we propose a novel algorithm based on deep reinforcement learning (DRL) for optimizing cost in multi-party computation software-defined security middle platforms (MPC-SDSmp) in real-time. To accomplish this, we first integrate fragmented security requirements and infrastructure into the MPC-SDSmp cloud model with privacy protection capabilities to reduce deployment costs. By leveraging the power of DRL and cloud computing technology, we enhance the real-time matching and dynamic adaptation capabilities of the security middle platform (Smp). This enables us to generate a real-time scheduling strategy for Smp resources that meet low-cost goals to reduce operating costs. Our experimental results demonstrate that the proposed method not only reduces the costs by 13.6% but also ensures load balancing, improves the quality-of-service (QoS) satisfaction by 18.7%, and reduces the average response time by 34.2%. Moreover, our solution is highly robust and better suited for real-time environments compared to the existing methods.

**Keywords:** software-defined security; deep reinforcement learning; cost optimization; Internet of Things; privacy protection





## 1. Introduction

In recent years, new infrastructure and digital transformation have enriched the variety of information access devices. The Internet of Things (IoT) [1], big data [2], edge computing, and machine learning technologies [3] are evolving rapidly [4,5]. The Internet is getting closer to people's lives, the risks to data are more complex and diverse, and the fragmentation of security operations is increasing. With the proliferation of IoT devices, vast amounts of data are being generated, and the number and types of these devices will continue to expand in the future. As a result, traditional IoT systems may not be equipped to adequately handle the associated challenges [6].

Fragmented security requirements and scenarios are significant challenges that Internet security has been faced with in recent years [7]. In addition, the mismatch between security assets and business scenarios is becoming more apparent [8]. In other words, as the variety and quantity of IoT devices and security infrastructure continue to increase rapidly, cost reduction has become the most pressing challenge for organizations. However, the mismatch between security measures and business scenarios presents a critical issue in cost optimization.

The Regulations on Security Protection of Critical Information Infrastructure highlight the security challenges and essential protection requirements facing critical information infrastructures. The traditional walled defense is not enough to cope with them, and it is necessary to build an active, proactive, resilient, and responsive security defense system with a security middle platform (Smp) as the core [8] to realize the protection concept

from security monitoring, the global situation, and capability planning to the orchestrated response.

Inspired by software-defined security (SDSec) and the security middle platform (Smp), the software-defined security middle platform architecture (SDSmp) [9] is built for the whole scenario, as shown in Figure 1. The purpose is to solve the problems of low utilization, difficult reuse of security resources, and high fragmentation of security requirements and scenarios to reduce costs. More importantly, the SDSmp provides a practical focus point to solve the mismatch problem between security protection means and business scenarios to improve the capability and flexibility of security protection.

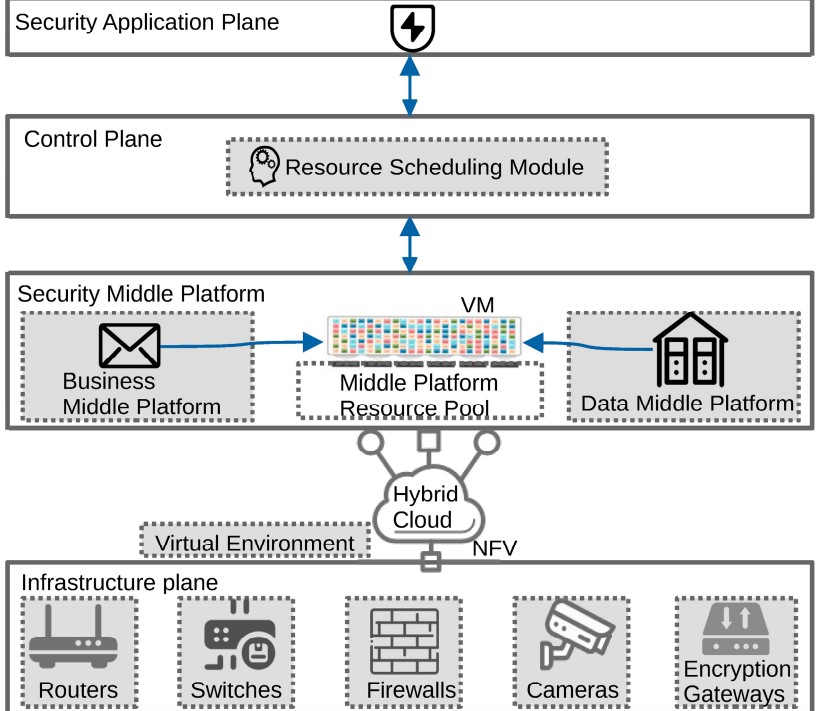

**Figure 1.** Software-defined security middle platform architecture (SDSmp).

SDSmp architecture enables the virtualization of the infrastructure layer through network functions virtualization (NFV) technology and cloud computing technologies such as IaaS, PaaS, and SaaS [9]. The Smp [7,8] has the advantages of both a data middle platform and a business middle platform, which can eliminate data silos, improve the resource reuse rate, and reduce development difficulty and cost. The resource scheduling module is the key to solving the mismatch problem between security protection and business scenarios.

The control plane scheduling module analyzes the required compute power based on the characteristics of the security services coming from the security application plane. The southbound application programming interfaces (API) allocates control to the available Smp resources for execution via the middle platform resource pool, which physically terminates at the infrastructure plane. Integrating all control components into the control plane improves security by allowing multiple security controls to be driven by the same data stream, and also reduces the overall cost of installing and configuring these controls and policies on each host [10].

In software-defined security, achieving cost optimization has been a persistent challenge. SDSec is distinguished by its manageability, dynamism, cost-effectiveness, and adaptability attributes. However, hardware-based security solutions for IoT devices are impractical due to their high costs and extensive deployment requirements [11,12]. Therefore, developing software-defined security technologies that can be easily configured in low-power IoT devices and offer the flexibility of timely upgrades is imperative [13].

Virtualizing security infrastructure reduces the need for physical deployment and is better suited to high-traffic scenarios. By optimizing and providing Smp resources on demand, virtual environments can significantly reduce the costs of security capital expenditure through resource and intermediate device optimization [14].

SDSmp provides security protection capabilities through security services, where user demands are transformed into security business requirements. Although the service of Smp resources greatly enhances developer productivity, it faces three primary challenges: privacy and security risks, information silos, and pricing mechanisms [15]. In the interconnected device-based IoT, many services collect potentially sensitive and private information of individual users due to differences in capacity, functionality, and security requirements of IoT devices, leading to a lack of user privacy protection [16]. Users are a critical component of the IoT [17], and they expect the security and privacy of their valuable data [18].

Multi-party computation (MPC) enables the secure computation of data while maintaining data privacy, which has significant potential in machine learning applications [19,20]. With the escalation in IoT terminal devices, the issue of data security is becoming increasingly prominent [21–23]. This paper presents a solution that lowers deployment costs through the SDSmp architecture and outlines an MPC method for a secure application plane that safeguards user privacy.

The Smp requires an appropriate scheduling policy to balance cost, resource reuse rate, load balancing, and security protection capabilities. Smp resources scheduling has not yet been studied, especially for real-time situations. The current protocols, including Azure IoT [24], are based on cloud computing and may not be able to meet the quality-of-service requirements of IoT systems. Given the real-time nature of network security attacks, improving the security protection of the SDSmp places higher demands on the quality of service, which is critical for security products with large security infrastructures and large numbers of user accesses in a fragmented landscape [25]. With today's increased challenges of fragmentation and mismatch, the fragmentation of security protection departments and vendors and the waste of resources due to security infrastructure deployed in multiple locations are severe [26]. The high cost makes it difficult for the security field to face a new security crisis. To address the above issues, a DRL-based algorithm for real-time cost optimization of a multi-party computation software-defined security middle platform (MPC-SDSmp) is proposed. A detailed algorithm design and implementation are provided, and a comprehensive performance evaluation is performed through extensive simulations of different types of workload scenarios.

To address the cost optimization challenges associated with software-defined security (SDSec), we have made significant efforts to reduce the cost of the SDSmp. The contributions can be summarized as follows:

- Architecturally, it reduces deployment costs by optimizing the architecture and increasing the reuse of security infrastructure resources. Specifically, SDSmp proposes an automated control architecture for fragmented security requirements and security scenarios, realizes real-time scheduling and automatic control of Smp resources, and makes the security infrastructure physically and geographically independent through NFV and cloud computing technologies. Multi-party computation (MPC) ensures that the security application layer is data agnostic and protects user privacy from leakage, enabling the security infrastructure to achieve resource reuse by building Smp.
- In terms of modeling, an SDSmp cost optimization model is established based on DRL algorithms so that the intelligent scheduler in the control plane can learn how to rationally select Smp resources based on real-time experience. This reduces operational costs and achieves high quality-of-service satisfaction, a low response time, and load balancing.
- An experimental SDSmp environment is built for implementation. The proposed DRL-based algorithm for real-time cost optimization of MPC-SDSmp is compared with existing real-time job-scheduling algorithms under different workload patterns. The

experimental results show that the proposed method outperforms existing real-time methods regarding cost, average response time, QoS satisfaction, and load balancing.

Although this article makes commendable contributions, it has limitations. One such limitation is that, like many other deep learning applications in various fields, the training process of the proposed method is conducted offline. While offline training can help minimize costs and prevent the consumption of valuable Smp resources, it also presents a limitation. However, once the model is trained, it can be deployed in real time during regular operations without additional offline training.

The rest of the paper is organized as follows: We report the related work in Section 2. We discuss the problem statement and introduce the proposed architecture model in Section 3. In Section 4, we formulate how intelligent job scheduling can be achieved with DRL. We present the performance evaluation of our algorithm in Section 5 and conclude this work in Section 6.

## 2. Related Work

Conventional approaches: In the past few decades, extensive research has been conducted on optimizing security infrastructure and IoT devices, and efforts are underway to enhance traditional methods. To address IoT heterogeneity, unify security infrastructure and IoT devices, decouple security operations and security control [27], and achieve unified management of security devices, various custom frameworks have been developed [10,16,27–32]. However, these frameworks are all based on cybernetic methods, which have limited performance improvements and are unsuitable for dynamic security scenarios. The round-robin methods in [29,33] suffer from severe delays and poor service quality, which is unacceptable for real-world security protection scenarios. Furthermore, the [30] method lacks the elasticity and scalability required to meet the needs of modern network security.

On the other hand, optimization algorithms based on linear programming and fixed strategies [34–37], as well as metaheuristics [38–45], have demonstrated their powerful capabilities in optimizing resource usage and job processing time. For instance, analyses on DDoS attacks in software-defined security (SDSec) [26,28,46] have achieved security protection through access control policies. However, the algorithm in [35] is strictly limited with limited applicability scenarios, and high-dimensional vectors learned from the source domain are unsuitable for the target domain. Likewise, the algorithm in [34] is strictly limited and unsuitable for highly dynamic security scenarios. While that in [36] constructs an anomaly detection module and a multi-level security response module to deal with various attacks, the physical infrastructure of security protection needs to be redesigned, and the control policies in the controller need to be reprogrammed, making it challenging to deploy. As for the algorithm in [37], security protection is implemented in a single kernel, with a narrow scope of application and easy to reach performance bottlenecks.

The heterogeneity of security infrastructure and IoT devices implies that deploying and configuring appropriate security mechanisms requires significant overhead. These methods often have strict limitations and cannot be used in different scenarios. Almost all conventional methods aim to address batch processing jobs, which are unsuitable for real-time, highly dynamic security capability services when processing transaction security middle platform (Smp) workloads, due to the huge overhead of solving optimization problems.

Particular methods have been developed to manage computing resources and jobs autonomously and interactively based on the state of security systems, such as the Monitor–Analyze–Plan–Execute (MAPE) loop [21,47–52]. Although monitoring the execution of security capabilities in software-defined network infrastructure is possible [48], it has limited functionality [49]. SDSec enhances the information security of vehicular ad hoc networks in large-scale wireless environments with high dynamic topology, but its applicability is limited. This method [50] is unsuitable for rapidly changing security environments due to poor controller interaction. Therefore, their planning phase still relies on solving

complex optimization problems. These problems have limited applicability and flexibility, making them unsuitable for highly dynamic and real-time security environments.

DRL-based methods: In contrast, deep reinforcement learning (DRL) methods have demonstrated high accuracy and the ability to handle complex control problems with high-dimensional state space and low-dimensional action space using deep neural networks [53]. DRL technology can effectively address complex decision-making problems [54–56], requiring only minimal training to solve various optimization problems [57]. Indeed, ref. [25] proposes a method for optimizing quality of service in the cloud and suggests that DRL-based algorithms are effective for cloud job scheduling in scenarios with variable workloads and complex decision-making.

Furthermore, reinforcement learning algorithms have been employed in other security-related fields to optimize routing and improve throughput [58–60], enhance the accuracy of multiclass classification tasks in intrusion detection [61], defend against distributed denial of service (DDoS) attacks in a software-defined network (SDN) [62], and improve system load balancing [63]. In contrast to these previous works, our research aims to use state-of-the-art DRL techniques to schedule heterogeneous security infrastructure and IoT devices to reduce deployment and operational costs. This represents a new area of research in SDSec and the IoT.

Efficient resource scheduling optimization is fundamental to improving the efficiency of highly dynamic, real-time heterogeneous security infrastructure. This presents a challenging task. The proposed algorithm based on DRL and multi-party computation software-defined security middle platforms (MPC-SDSmp) aims to address this issue. The algorithm optimizes resource scheduling to improve the utilization of Smp resources and reduce operating costs while protecting user privacy and ensuring security capabilities. Overall, the algorithm provides a comprehensive solution for cost optimization and security enhancement in the IoT and SDSec.

## 3. Our Scheduling Model

For the SDSec cost optimization problem, from the SDSmp in Figure 1, the MPC-SDSmp cost optimization architecture based on DRL was designed as shown in Figure 2. The essence of separating the software-defined [64] control plane and the data plane is to unify the control plane scheduling for Smp resources in the virtualized resource pool [18]. On the one hand, it saves costs by avoiding duplicate infrastructure deployment everywhere. On the other hand, optimizing resource scheduling during operation ensures performance and reduces costs. Smp, the purpose of a sizeable middle platform and a small front platform [9], fragmented security requirements, and security scenarios place higher demands on resource scheduling. Different control plane scheduling algorithms significantly impact the performance of SDSmp.

First, in the northernmost security application layer, as shown in Figure 2, users access the system through the foreground application provided by the security application layer. For SDSec to achieve cloud deployment and user privacy protection, MPC is required for user data in the foreground application layer [19]. Each user submits data before local secret generation, as (A) shown in Figure 2, such as homomorphic encryption [20]. Followed by the grouping of similar terminal geographical locations, similar security business users form a group, as (B) in Figure 2, leading to secret sharing and exchange. In addition, as (C) in Figure 2, there is local aggregation by group, and the request of the group with lower relevance to the user is submitted to the corresponding type of security business in the security application layer.

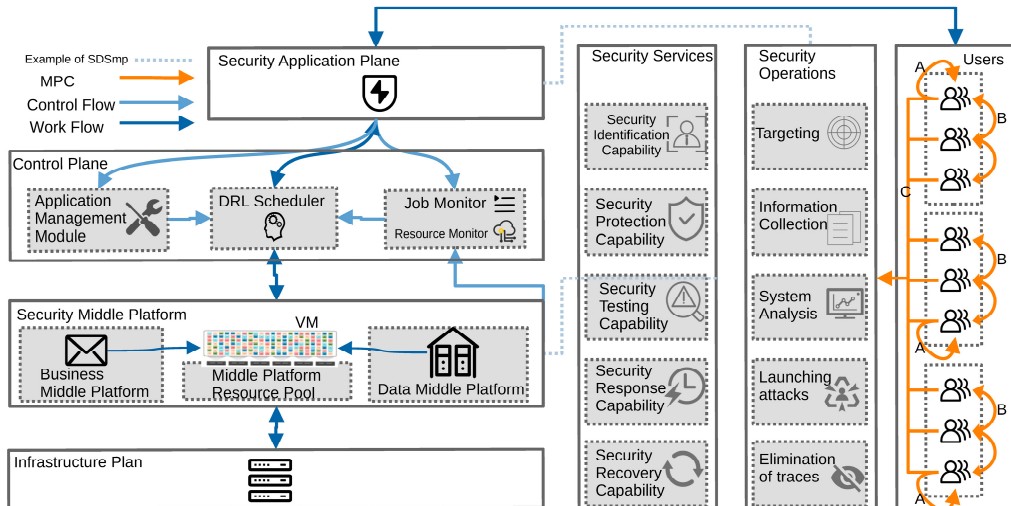

**Figure 2.** MPC-SDSmp cost optimization architecture based on DRL.

In addition, the foreground security business requests Smp resources in the form of jobs, and the resources provide the appropriate security in the form of capabilities. Smp resources encapsulate a specific security infrastructure, eliminating duplicate deployments of traditional security defenses to reduce costs and ensure user privacy and security. With the addition of cloud deployments, it is critical that security services can be successfully provisioned to the right security business. Properly scheduling jobs to the appropriate Smp resources is essential to solving this problem.

The control plane of MPC-SDSmp is with the security application plane on the north side and the security middle plane on the south side. The optimization architecture is shown in Figure 2, with three arrows to distinguish between the control information and the actual working information during transmission, and the MPC process before the security application plane is connected to the control plane. A dashed line indicates a specific example of an Smp. The MPC-SDSmp cost optimization architecture consists of users, a security application plane, a control plane, a security middle platform (Smp), and an infrastructure plane. The critical part of the control plane for scheduling is the DRL scheduler of the resource scheduling module. Other key components, such as job queues, application management modules, and information collectors, include resource and job monitors that collect information about Smp resources and foreground jobs in the middle platform resource pool.

In actual operation, users continuously submit appropriate security business requirements to the foreground using endpoint security products in the security application plane while protecting user privacy through the above MPC process before submission. The goal of the control plane is to realize the docking of security business and security services, and release the security protection capability by scheduling appropriate security business foreground jobs to Smp resources of complementary capabilities.

Reusing resources and providing capabilities in the form of services is the main thrust of the Smp. Take a network security defense as an example, as shown in Figure 2. The Smp abstracts the capabilities into line-by-line services, and Smp resources of the complementary capabilities provide the services. The requests from security applications are first transformed into the appropriate class-by-class security operations. Then, the various security services request all required services from the middle platform resource pool as job requests.

Specifically, various security application requests are first transformed into security operations by performing parallel classification and refinement at the security application layer. The security operations submit requests for simple jobs that are highly decoupled, relatively low demand, and fine-grained. Jobs are assigned to equally fine-grained Smp resources during scheduling, and they complete the execution of each job in the form

of provisioned services. The final unified assembly of jobs improves parallelism and largely avoids the problems caused by logical dependencies, predecessor and successor relationships, and resource contention between traditional jobs.

The application management module on the control plane analyzes foreground jobs from the security business and job attributes such as resource utilization, compute power, memory, required response time, and QoS. The virtualized Smp is deployed in the public cloud to maximize the benefits of the SDSmp. Smp resources are modeled and encapsulated into virtual machines (VMs) using IaaS, PaaS, and SaaS technologies.

The job-scheduling process for a complete middle platform pool is as follows. First, the security business in the application layer completes the MPC and submits the job. Immediately after that, the application management module analyzes the job type. Then, the scheduler in the control plane resource scheduling module searches for an appropriate encapsulated Smp resource virtual machine (VM) in the middle platform resource pool to assign and execute the job to provide the required security services. Thus, the job scheduler is the core module that makes decisions based on the QoS requirements of the security service at a given time interval to make it as cost-effective as possible. In the operating mechanism corresponding to DRL, the job scheduler assigns a foreground job to a particular virtual machine in the Smp resource pool, on the basis of which the environment provides rewards and updates the state, iteratively achieving intelligent learning of the scheduler. In this process, resource and job monitors are responsible for managing the workload and performance of the job queue, as well as the execution and assignment of jobs.

To model the optimization problem, the mathematical definitions of the foreground job load and Smp resources are given below, along with the execution mechanism for scheduling, using the notation shown in Table 1.

**Table 1.** Notations used in our scheduling model.

| Notation | Meaning |
|---|---|
| $J^{id}$ | The ID of the foreground job |
| $J^{at}$ | The arrival time of the foreground job |
| $J^t$ | The type of the foreground job |
| $J^l$ | The length of the foreground job |
| $J^q$ | The QoS requirement of the foreground job |
| $J^{rt}$ | The response time of the foreground job |
| $J^{et}$ | The runtime of the foreground job |
| $J^{wt}$ | The waiting time of the foreground job |
| $J^{\cos t}$ | The cost of the foreground job |
| $V^{id}$ | The ID of Smp resource (VM) |
| $V^t$ | The type of Smp resource (VM) |
| $V^p_{com}$ | The computing processing speed of Smp resource (VM) |
| $V^p_{io}$ | The IO processing speed of Smp resource (e.g., instructions per second) |
| $V^{it}$ | The available time of Smp resource (VM) |
| $\cos t^{VMe}$ | The execution cost of Smp resource (VM) per time unit |
| $\cos t^{VMs}$ | The start-up cost of Smp resource (VM) |
| $\mathbb{R}^{QoS}$ | The reward function reflecting QoS satisfaction |
| $\mathbb{R}^{\cos t}$ | The reward function reflecting user satisfaction with costs |
| $\mathbb{R}$ | The reward function of DRL |
| $Sat$ | Whether security operations are successfully dispatched and security protections take effect |

### 3.1. Foreground Job Characteristics

Security operations call security services as foreground jobs. The design of security operations and security services dramatically reduces the coupling, correlation, and dependency between jobs. Furthermore, since this paper focuses on using DRL to handle security operations that require a real-time response in the SDSec domain, the cost of SDSec is reduced in an automated, real-time, cost-aware manner. Based on the above, it can be

assumed that the operations in the real-time scenario are independent of each other, and operations cannot interfere with each other during execution. To reduce the myriad of possible actions in the DRL, an event-driven decision mechanism is introduced that analyzes the foreground job in real-time as soon as it reaches the control plane. This information is used to train the job-scheduling mechanism. For the proposed model, two typical types of jobs are considered: compute-intensive jobs and I/O-intensive jobs. The following parameters are modeled for the jobs coming from the foreground security operations:

$$\mathbb{J}_i = \left\{ J_i^{id}, J_i^{at}, J_i^t, J_i^l, J_i^q \right\} \tag{1}$$

where $J_i^{id}$ is the foreground job ID, $J_i^{at}$ is the job arrival time, $J_i^t$ is the job type, $J_i^l$ is the job length, and $J_i^q$ is the QoS requirement of the job (fixed length period). Moreover, $J_i^q$ reflects the foreground service's expected completion time and security level requirement.

### 3.2. Security Middle Platform Resources

Security middle platform (Smp) resources provide protection capabilities for security operations through security services. Virtualized Smp is deployed in the public cloud to maximize the benefits of a cloud-based, software-defined security middle platform (SDSmp). Smp resources are encapsulated in virtual machines (VMs) using IaaS, PaaS, and SaaS technology models. In the SDSmp job-scheduling model, Smp resources, which correspond to clusters of virtual machines (VMs), are the logical execution units. The actual physical execution unit is the specific infrastructure plane security appliance. The infrastructure plane is functionally mapped to different VM clusters through NFV technology and cloud computing [65] to achieve logical device independence.

When scheduling jobs, because the jobs submitted by the foreground security operations may differ, they have different response times for execution on different types of Smp VMs. Similar to the job load, consider two types of Smp resources: I/O-intensive $VM_{t1}$, which connects to data-intensive devices such as monitors at the infrastructure layer, and compute-intensive $VM_{t2}$, which connects to compute-intensive devices such as data encryption and decryption modules at the infrastructure layer. Each Smp resource is defined as:

$$\mathbb{V}_j = \left\{ V_j^{id}, V_j^t, V_{com_j}^p, V_{io_j}^p, \cos t_j^{VMe}, \cos t_j^{VMs} \right\} \tag{2}$$

where $V_j^{id}$ is the Smp resource ID, $V_j^t$ is the Smp resource type, $V_{com_j}^p$ is the computing speed of the Smp resource, and $V_{io_j}^p$ is the IO speed of the Smp resource. In addition, $\cos t_j^{VMe}$ is the hourly cost of using or renting Smp resources, which differs for different types of security middle platform resources. $\cos t_j^{VMs}$ is the start-up cost of the Smp resource (VM), which is negligible.

### 3.3. Job-Scheduling Mechanism

After the scheduling decision, when a job is assigned to a particular Smp VM instance, the job enters a wait queue, $L_j^i$. Without loss of generality, it is assumed that each VM instance can exclusively execute only one job in its wait queue at any given time. The job scheduler is the core component that assigns jobs to Smp resources in the appropriate pool of Smp resources on the basis of the requirements of the security business. If the wait queue is empty, the assigned job will smoothly pass through the queue to the virtual machine and be executed immediately; otherwise, it will enter the wait state first. According to the above assumptions, the response time of a job consists of two parts: wait time and execution time, and the response time can be expressed as:

$$J_i^{rt} = J_i^{et} + J_i^{wt} \tag{3}$$

where $J_i^{rt}$ is the job response time, $J_i^{et}$ is the job execution time, and $J_i^{wt}$ is the job wait time. The job execution time varies depending on the scheduling of different types of Smp

resources. As mentioned earlier, the job transfer time for an exact fixed type of foreground job is negligible because each Smp resource operates in parallel [66]. Furthermore, the main factor affecting the job execution time on the middle platform resource is the length corresponding to that job type; the length of other types of jobs is truncated in comparison, can be ignored, and has no practical impact. Therefore, the job execution time is defined as:

$$J_i^{et} = max(\frac{J_{com_i}^l}{V_{com_j}^p}, \frac{J_{io_i}^l}{V_{io_j}^p}) \tag{4}$$

where $J_i^{et}$ is the job execution time, $J_{com_i}^l$ is the required computation length of the job, and $J_{io_i}^l$ is the required IO length. $V_{com_j}^p$ is the computation processing speed of the Smp resource, and $V_{io_j}^p$ is the IO processing speed of the Smp resource. The job type that corresponds to the length is the primary influencing factor. However, a job can be scheduled to a matching or different type of Smp resource, similar to Cannikin Law [67]. If the job type matches the resource type, the job execution time is short due to the strong performance of the corresponding type of Smp resource. If it does not match, the job execution time is much longer due to the weak performance of the corresponding type of Smp resource. In addition, the job wait time affects resource scheduling, and the wait time is defined as follows:

$$J_i^{wt} = \begin{cases} 0, \text{if } L_j^i = 0 \\ \sum_{n=0}^i J_n^{et}, \text{else} \end{cases} \tag{5}$$

where $J_i^{wt}$ is the job wait time. If the wait queue is empty, the job is executed immediately. Otherwise, it must wait. The wait time is the sum of the execution times of all the previously arrived jobs. $J_i^{et}$ is the job execution time. When the foreground job $\mathbb{J}_i$ is scheduled for resource $\mathbb{V}_j$ and completes processing, the free time of the Smp resource is updated as follows:

$$V_j^{it} = J_i^{wt} + J_i^{at} + J_i^{et} \tag{6}$$

Smp resources enable the SDSmp in the form of services and provide appropriate security protection for security services in foreground job execution. A proper balance of cost, load balancing, and QoS is necessary for the Smp and scheduling systems. The SDSmp allows end users to specify QoS requirements on the basis of security protection levels when submitting foreground job requests. Security services often have strict deadline response time requirements, especially in real-time environments. The QoS requirements for security services are defined as the maximum acceptable response time for a foreground job. QoS satisfaction is defined by the following formula to determine the success of scheduling each foreground job:

$$Sat_{ij} = \begin{cases} 1, J_i^{rt} \leqslant J_i^q \\ 0, J_i^{rt} > J_i^q \end{cases} \tag{7}$$

Each foreground job and service request of the security service has a different execution deadline (expectation) depending on the security protection level. Assume that the execution result of the Smp resource can be returned within the deadline. In this case, the scheduling execution is successful, the QoS requirements are met, and the security protection capability takes effect. Otherwise, the job request is canceled, and this scheduling execution fails. For the security service to execute successfully, contacting the front office and re-executing the alternate job after updating the QoS requirements due to the scheduling failure is necessary. Foreground job costs are defined as follows:

$$J_i^{\cos t} = J_i^{et} \times \cos t_j^{VMe} + \cos t_j^{VMs} \tag{8}$$

where $J_i^{\cos t}$ is the execution cost of the job, and $J_i^{et}$ is the actual execution time of the job on the Smp resource. $\cos t_j^{VMe}$ is the usage cost of the target Smp resource to which the job is

dispatched, and $\cos t_j^{VMs}$ is the startup cost of the target Smp resource. Since Smp resources are deployed in the cloud, they are occupied when scheduled and released immediately after use, and each startup time is short. The data transfer and job execution are parallel to the design of the wait queue, and the transfer time is negligible compared to the run time [66]. Thus, only the run-time cost needs to be considered in the actual job operation cost.

## 4. Methodology

As shown in Figure 2, in the SDSmp, Smp resources of different capabilities provide different services, and all these capabilities constitute various security protection means. Furthermore, specific security business scenarios correspond to specific security applications, which submit jobs in the form of security business to apply for the services of the Smp.

Security research has identified various fragmented security infrastructure functions spread across multiple locations and with many redundancies. How to efficiently utilize them to reduce redundancy and cost has always been a problem. The purpose of creating reusable and geo-physically independent Smp resource pools is to address the challenge of security infrastructure fragmentation and reuse. The proliferation of new fragmentation attacks is creating a dizzying array of security protection requirements, and individual companies and organizations are acting like isolated information silos [15], repeatedly building wheels. Therefore, all kinds of security operations are designed in the SDSmp. The development and deployment of security applications in the security application layer, similar to the portal that provides users with multiple security protection means, solve the problem of repeated development and fragmentation.

In this paper, the MPC-SDSmp is used to reduce the deployment cost. Then, we reduce the operating cost of Smp by automating the real-time cost awareness and ensuring high-security service QoS satisfaction and low load balancing rate. Finally, a reliable and low-cost real-time scheduling policy is generated for Smp resources. The QoS scheduling is considered successful if each security service corresponding to the security protection completes the service within the expected time, as shown in Equation (7). If the services are completed within the expected time, the scheduling is considered successful, the security protection fails, and the job requirements can be satisfied. The quality-of-service requirements for security services are positively correlated with the security level. The higher the level, the higher the real-time responsiveness required for security services and the tighter the time constraints. In addition, security business quality-of-service requirement is negatively correlated with cost and load balancing, which are in direct conflict. The higher the QoS requirement, the tighter the deadline, the greater the number of redundant Smp resources required, and the higher the load balancing rate.

Cost optimization in multivariate real-time environments is a significant challenge in SDSec because the fragmentation and mismatch between security protection means and business scenarios lead to difficulties applying mainstream cost optimization schemes and performance degradation. Traditional cybernetic and heuristic-based scheduling algorithms are difficult to apply, so we propose a DRL-based algorithm for real-time cost optimization of MPC-SDSmp. In addition, the model training phase is performed offline, and the operational decision phase is performed online, which means only valuable common Smp resources are occupied and the algorithm can be better adapted to changing security scenarios.

For the sake of clarity, the definitions of all symbols used in our DRL-based algorithm are given in Table 2.

**Table 2.** Notations used in our DRL-based scheduling.

| Notation | Meaning |
|---|---|
| $\mathbb{R}^{QoS}$ | The reward function reflecting QoS satisfaction |
| $J^{et}$ | The runtime of the foreground job |
| $J^{rt}$ | The response time of the foreground job |
| $J^q$ | The QoS requirement of the foreground job |
| $\mathbb{R}^{\cos t}$ | The reward function reflecting user satisfaction with costs |
| $\lambda$ | The hyperparameter is used to indicate the maximum cost $\max(J^{\cos t})$ of the job |
| $J^{\cos t}$ | The cost of the foreground job |
| $\mathbb{R}$ | The reward function of DRL |
| $\mathbb{R}uturn$ | The return function of DRL |
| $\gamma$ | The discount factor of DRL |
| $Q$ | The Q-value function of DRL |
| $\mathbb{A}$ | Action space |
| $\mathbb{S}$ | State space |
| $t$ | The current time |
| $\theta$ | The random parameters of Q |
| $B$ | The training minibatch |
| $\phi\phi^{target}$ | Fixed parameters when calculating the MSE loss |
| $\epsilon$ | The exploration rate |
| $f$ | The learning frequency |
| $S_\Delta$ | The minibatch |
| $\eta$ | The replay period |
| $\Delta$ | The replay memory |

### 4.1. Basics of DRL

Deep Q-learning (DQN) is a model-free reinforcement learning (RL) [53] algorithm where the agent requires little input of a priori knowledge. The reinforcement learning model consists of environment, agent, action, state, reward, and a value function, $Q : \mathbb{S} \times \mathbb{A} \Rightarrow \mathbb{R}$, that aims to predict the action that maximizes the reward. The return function, $\mathbb{R}uturn$, is based on the reward function. The agent makes decisions through trial-and-error interactions, and after each executed action, the environment moves to the next new state, $\mathbb{S}_{t+1}$. At the same time, the agent receives a reward, $\mathbb{R}_t$. The experimental replay mechanism is continuous [25].

$$\mathbb{R}uturn = \sum_{t=0}^{n} \mathbb{R}_t \gamma^t \qquad (9)$$

where $\gamma$ is the discount factor that weights future rewards to guide whether the model focuses more on current or on possible future rewards, and $\mathbb{R}uturn$ is a weighted accumulation of all $\mathbb{R}_t$ from start to finish. The most common loss used for training is the mean squared error (MSE) loss, which can be expressed as:

$$min_\phi \sum_{i=1}^{|B|} (r_{t_i} + \gamma max_{\hat{a} \in \mathcal{A}} Q_{\phi^{target}}(s_{t_i+1}, \hat{a}) - Q_\phi(s_{t_i}, a_{t_i}))^2 \qquad (10)$$

where $B$ is the training minibatch and $\phi^{target}$ is fixed when calculating the MSE loss, $r_t$ is the reward obtained by taking action for state $s_t$, and $\gamma$ is the discount factor, and its value lies in (0,1). The agent uses the rewards generated by the deep neural network (DNN) to feedback to the environment and makes decisions about specific states. All state–action pairs are correlated.

### 4.2. Our DRL-Based Scheduling

As shown in Figure 3, two types of arrows are used to indicate the data transmitted and the functions represented. Cost marked in red is the main optimization objective. The SDSmp control plane schedules the foreground jobs from the security application plane's

security operations. First, the application management module cleanses and categorizes them into different job types. Agents in the resource scheduling module coordinate with the resource pool management module to assign each job to an Smp resource in the resource pool of the most appropriate type. Smp resources provide automized security services that provide appropriate security capabilities to foreground jobs. The final job is executed in the encapsulated, mapped security infrastructure layer.

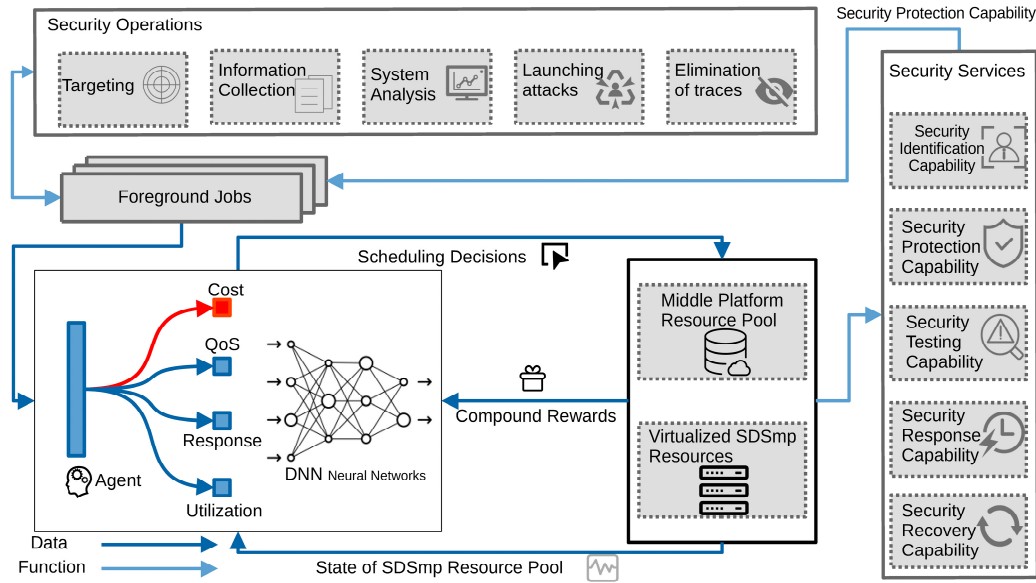

**Figure 3.** A DRL-based algorithm for real-time cost optimization of the MPC-SDSmp.

In an Smp environment, the characteristics and types of incoming workloads are unpredictable. RL-based models perform well in such variable scenarios because they require minimal input of a priori experience, such as state shifts and other system information. In each decision iteration, the RL agent observes the current state of the environment and then uses DNN to estimate the Q-values of all Smp resources available in the pool. It also trains itself to improve future decisions.

Depending on the policy, an instance is selected from the Smp resources pool to perform the job and receive the reward. Due to the ample state space, the training time of the DNN can also be considerable. To avoid this situation, an event-driven decision mechanism is used. Based on this mechanism, the intelligent agent of the resource scheduling module makes a real-time decision when a new job arrives. All jobs follow the first-come, first-served (FCFS) principle. When a job arrives, it must be assigned to the SDSmp resource pool. The real-time decision mechanism also reduces the number of optional actions. The proposed approach is divided into two phases: decision and training, as described below.

The deep Q-learning technique assigns jobs to appropriate VM instances of Smp resources. Decisions are made based on specific requirements, and agents are rewarded accordingly. The agent checks to update the current state of the environment to make the next possible decision. The following are the critical components of the reinforcement learning model.

### 4.2.1. Action Space

Action space ($\mathbb{A}$) is the set of actions that an agent can perform in a given environment [53]. Action space can be represented as the set of the total number of Smp resource VM instances in all resource pools described by $\mathbb{A}$, and includes assigning a foreground security operation to a particular instance in the Smp resources pool. The length can be represented as the number of all available Smp resources [25]. Each virtual machine has

its queue to hold incoming job requests. There is no limit to the length of incoming job requests.

$$\mathbb{A} = \{ a_{t_i} | a_{t_i} \in VM_{t1} \cup VM_{t2} \} \tag{11}$$

where $VM_{t1}$ and $VM_{t2}$ refer to different types of Smp resources to provide security services to different types of foreground jobs, for example, set $VM_{t1}$ as a high-CPU-type Smp resource and $VM_{t2}$ as a high-I/O-type Smp resource.

### 4.2.2. State Space

State space ($\mathbb{S}$) is the set of all possible states that the agent updates based on actions that generate a finite state space [53]. For an SDSmp, a new foreground security service submission of job *i* arrives at time *t*. The overall state of Smp resources and the job's current state can describe the state space at that time.

$$\mathbb{S} = \left\{ s_{t_i} | s_{t_i} \in S_{job} \cup S_{VM} \right\} \tag{12}$$

where $S_{VM}$ is the state of all Smp resources for the job *i* arriving at time *t*, and $S_{job}$ is the current state of jobs to schedule.

### 4.2.3. Action Selection and State Transition

Our model takes actions considering the current state and the predicted future state from the Q-value from the DNN. Firstly, the algorithm allocates the jobs randomly on the Smp resource VM for which the probability would be $\epsilon$, and the value of $\epsilon$ gradually decreases as the algorithm learns. The agent randomly allocates the jobs and explores several possibilities with the greedy policy. The highest predicted Q-value would be selected here. As the jobs are allocated, the state changes. The state will transfer from $\mathbb{S}_t$ to $\mathbb{S}_{t+1}$.

### 4.2.4. Reward Function

Reward function ($\mathbb{R}_i$). After acting on the current state, $\mathbb{S}_t$, the system updates to state $\mathbb{S}_{t+1}$ and receives a reward, $\mathbb{R}_i$, from the environment. In each iteration, the environment gives a reward. The reward is positive or negative, depending on the action. The agent can receive different rewards for actions, and the reward function guides the agent to make intelligent decisions for the goals of the job-scheduling framework. In this model, the optimization goal of job scheduling is low cost with high QoS. Therefore, the reward function $\mathbb{R}_i$ consists of two components, $\mathbb{R}_i^{\cos t}$ and $\mathbb{R}_i^{QoS}$. The lower the cost, the higher the $\mathbb{R}_i^{\cos t}$, and the higher the QoS satisfaction, the higher the $\mathbb{R}_i^{QoS}$. The execution failure for $\mathbb{R}_i$ is 0. For each job that satisfies the QoS requirements, the smaller the response time and the higher the service quality satisfaction. Based on this, the QoS reward for a job is defined as follows:

$$\mathbb{R}_i^{QoS} = \begin{cases} \frac{J_i^{et}}{J_i^{rt}}, J_i^{rt} \leqslant J_i^q \\ 0, J_i^{rt} > J_i^q \end{cases} \tag{13}$$

where $J_i^{rt}$ is the job response time, $J_i^q$ is the QoS requirement time, and $J_i^{et}$ is the job execution time. If, and only if, the job response time is less than the QoS requirement time, the scheduling meets the QoS requirement, and the scheduling succeeds; otherwise, it fails. This is critical to the security business because a failure of the foreground job scheduling means that Smp resources cannot successfully provide the security protection capability to the security business within the specified time. For the security service to run successfully, it must report back to the foreground and re-execute the alternate job after updating the QoS requirements due to the scheduling failure.

$$\mathbb{R}_i^{\cos t} = -\frac{2}{\pi} \arctan(J_i^{\cos t} - \lambda) \tag{14}$$

where $J_i^{\cos t}$ is the job execution cost and $\lambda$ is the hyperparameter representing the maximum cost, $\max(J_i^{\cos t})$, of the job. As shown in Figure 4, the image of the reward function, $\mathbb{R}_i^{\cos t}$, decreases nonlinearly with the job cost, $J_i^{\cos t}$. When the cost tends to 0, the user is more tolerant of the cost change, and $\mathbb{R}_i^{\cos t}$ changes very slowly. When $\max(J_i^{\cos t})$ cost tends to be maximum, $\frac{J_i^{\cos t}}{\lambda}$ tends to be 1, and the user can hardly accept the high price, and $\mathbb{R}_i^{\cos t}$ drops to 0 quickly.

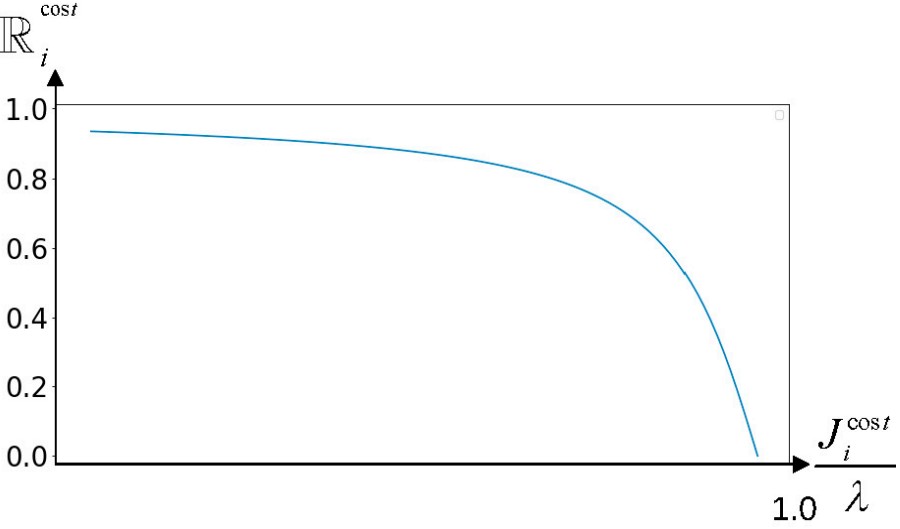

**Figure 4.** Nonlinear decrease of the reward function $R^{cost}$ with $J^{cost}$.

The reward function, $\mathbb{R}_i$, is defined as follows:

$$\mathbb{R}_i = \mathbb{R}_i^{\cos t} \times \mathbb{R}_i^{QoS} \tag{15}$$

If $\mathbb{R}_i^{QoS} = 0$, scheduling fails and $\mathbb{R}_i$ is 0; otherwise, scheduling succeeds. $\mathbb{R}_i^{QoS}$ corresponds to QoS satisfaction. The higher the satisfaction is, the higher $\mathbb{R}_i^{QoS}$ and $\mathbb{R}_i$ are. $\mathbb{R}_i^{\cos t}$ responds to user satisfaction with the price. The lower the cost is, the higher $\mathbb{R}_i^{\cos t}$ and $\mathbb{R}_i$ are, and the trend of $\mathbb{R}_i^{\cos t}$ is shown in Figure 4.

### 4.3. Training Phase

The details are shown in Algorithm 1. To learn from experiences, DRL stores the transition values of the current state, action, reward, and next state in the replay memory $\Delta$ with capacity $N_\Delta$. Moreover, the parameter $\theta$ of the DQN is updated using a minibatch data set that contains $S_\Delta$ samples randomly chosen from replay memory $\Delta$. The storage time is after each U decision set to avoid excessive time complexity, U > 1. The experience replay mechanism learns from random samples, which reduces data correlation and reduces the variance of $\theta$. Q-values are generated using the target network, and the divergence and oscillations of the DNN are eliminated using the target network and the evaluation network, which have the same structure but different parameters [55].

As with deep learning applications in other domains, the training process of the proposed method is performed offline, which maximizes cost savings and avoids tying up valuable Smp resources. Once the model is trained, it can be scheduled in real time and does not need to be trained offline again in subsequent normal operations. Specifically, the hidden layer of the model uses 20 neurons, and the overhead is close to 0 when the model is not large, while the scheduling time is always less than 10 ms, which is practically negligible.

---

**Algorithm 1:** The DRL-based algorithm for real-time cost optimization of the MPC-SDSmp

| | |
|---|---|
| 1: | Input: initial $\epsilon$, $\alpha$, $\gamma$, learning frequency $f$, start learning time $\tau$, minibatch $S_\Delta$, replay period $\eta$ |
| 2: | Initialize replay memory $\Delta$ with capacity $N_\Delta$ |
| 3: | Initialize evaluation value function $Q$ with random parameters $\theta$ |
| 4: | Initialize target value function $\hat{Q}$ with random parameters $\theta'$ |
| 5: | for each new job $j$ arrives at $t_j$ do |
| 6: |   with probability $\epsilon$ randomly choose an action; otherwise $A_j = argmax_A Q(S_j, A; \theta)$ |
| 7: |   Schedule job $j$ according to action $A_j$, receive reward $R_j$, and observe state transition at next decision time $t_{j+1}$ with a new state $S_{j+1}$ |
| 8: |   Store transition $(S_j, A_{j+1}, R_{j+1}, S_{j+1})$ in $\Delta$ |
| 9: |   if $j \geqslant t$ and $j \equiv 0 \ mod \ f$ then |
| 10: |     if $j \equiv 0 \ mod \ \eta$ then |
| 11: |       Reset $\hat{Q} = Q$ |
| 12: |     end if |
| 13: |     randomly select samples $S_\Delta$ from $N_\Delta$ |
| 14: |     for each transition $(S_j, A_{j+1}, R_{j+1}, S_{j+1})$ in $S_\Delta$ do |
| 15: |       $target_k = r_k + \gamma max_{A'} \hat{Q}(S_k + 1, A'; \theta')$ |
| 16: |       update DNN parameters $\theta$ with a loss function of $target_k - Q(S_k, A_k; \theta)^2$ |
| 17: |     end for |
| 18: |     Gradually decrease $\epsilon$ until to the lower bound |
| 19: |   end if |
| 20: | end for |

---

## 5. Evaluation

A series of experiments were conducted to evaluate the proposed DRL-based real-time cost optimization algorithm of the MPC-SDSmp and to compare it with five standard real-time job-scheduling methods. First, the experiments were reasonably set up and simplified as necessary to ensure they were smooth and convincing. Then, the proposed model and parameters in the comparison method were illustrated, the five control methods and each parameter description were introduced, and then three different workload models were set up to simulate the actual situation. Good simulation experiments were conducted to verify that the proposed algorithm can adapt to different types of environments. In the experimental results of each set of simulated environments, the reader can see the advantages and improvements of the proposed method for the SDSmp. Long-term experiments were also conducted to demonstrate the performance of the algorithm further. The experimental hardware–software configuration was Python3 (Python Software Foundation), TensorFlow (Google), using a 2.7 GHz Intel Core i5 processor and a machine (DELL) with 16 GB of RAM.

### 5.1. Experimental Framework

Consider the Smp resources pool that has been pooled and virtualized on the Smp, shown as different types of application programming interfaces (API) with different performances invoked uniformly for the resource pool management module on the control plane. To simplify the experiment, Smp resources of the pool were set to be of high-CPU type and high-I/O type. The jobs passed to the control plane from the application plane application management module were always compute-intensive and I/O-intensive.

The control plane schedules jobs from the northbound application plane to the Smp for execution. Security business foreground jobs run fast when scheduled on the same type of Smp resources and run slowly when running different types of jobs. The average processing capacity of Smp resources to handle different types of jobs is shown in Table 3.

**Table 3.** Average processing capacity of security middle platform resources.

|  | Computing-Intensive Job | IO-Intensive Job |
|---|---|---|
| High-CPU Smp resource | AVG 1000 MIPS STD 100 MIPS | AVG 500 MIPS STD 50 MIPS |
| High-IO Smp resource | AVG 500 MIPS STD 50 MIPS | AVG 1000 MIPS STD 100 MIPS |

In the experiments, job lengths were generated by default from a normal distribution with a mean of 100 MIPS and a standard deviation of 20 MIPS. Each job's QoS requirements (i.e., acceptable maximum response time) were generated uniformly and randomly between 250 ms and 350 ms. New arrival job types were chosen uniformly and randomly between compute-intensive and IO-intensive types. The probability distributions of job arrival rates and job types were refreshed every 5 s in a cycle. For each simulated workload pattern, the experiments randomly generated 20 instances of safe middle resource VMs and tracked each safe resource for 300 s from start to finish of operation.

Moreover, the MPC-SDSmp cost optimization architecture based on DRL uses a feedforward neural network to construct the underlying DNN, which has a fully connected hidden layer with 20 neurons. We set the capacity of reply memory $N_\Delta$ to be 1000, and the size of the minibatch, which helped to reduce the correlation in data $S_\Delta$ to be 40. The AdamOptimizer algorithm was used to evaluate the network parameters with a learning rate of 0.01. Moreover, parameters were cloned from the evaluation network to the target network every 50 decision sets. After enough transition samples were accumulated in the replay memory, the DNN started training. We set $\tau = 500$, $f = 1$, $\gamma = 0.9$, and $\epsilon$ to be decreased from 0.9 by 0.002 in each learning iteration.

*5.2. Baseline Solutions*

To evaluate the performance of the proposed MPC-SDSmp cost optimization architecture based on DRL (denoted as DQN), we compared it with five other standard methods: random scheduling method [28,30], round-robin scheduling method [29,33], earliest scheduling method [10,16,31,32], suitable scheduling method [42,45], and sensible scheduling method [44].

Among the standard cybernetic scheduling algorithms, the random scheduling method [28,30] is straightforward and chooses a random VM instance for each job. The round-robin scheduling approach focuses primarily on scheduling jobs to VM instances. As a result, VM instances are selected in a round-robin [29,33] order to execute incoming jobs. The earliest scheduling method [10,16,31,32] is a first-come, first-served policy in which newly arriving jobs are scheduled on the earliest available VM instance.

The suitable scheduling method [42,45] is a greedy algorithm that tries to make the best choice. Unlike the earliest scheduling methods, the suitable scheduling method considers two factors, the time factor and whether the type of the selected VM instance matches the type of the newly arrived job. It always reduces execution time by finding the local optimum, not the overall optimum, and by assigning the job to the correct type of VM instance. This means the suitable scheduling method assigns newly arrived jobs to the first busy VM instance of the correct type.

The sensible scheduling method [44] is an adaptive heuristic algorithm that uses a random routing policy based on the expected QoS, i.e., the average job response time. Jobs are assigned to VM instances with a higher probability of a lower average response time. The sensible scheduling method requires a continuous observation time D and a discount factor a. The experimental settings were D = 5 s, a = 0.7, and D = 0.2 s, a = 0.7.

We used four different metrics to evaluate the performance of each method [25]. The first metric was QoS satisfaction, which measures how many jobs are completed. This scheduling satisfies the QoS requirements, and the scheduling is successful if and only if the response time of a job is less than the predefined QoS requirements. The second metric was average response time, which measures the average time it takes to process

each job. The third metric was cost, which measures the cost of operating all Smp resources. The fourth metric was the load balancing rate, which measures Smp resources utilization. Generally, the lower the load balancing rate is, the better the scheduling method is. In other words, to handle jobs of the same intensity, an efficient scheduling method will use fewer resources in the scheduling process and ultimately have a lower load balancing rate.

In addition, we set up three different workload patterns [25], and the job arrival rates of the workloads were randomly generated according to a regular pattern, as summarized below in Table 4, showing the parameters of the three simulated workload patterns of the experimental environment. The probability distribution of the job types changed continuously with time.

**Table 4.** Generation of load modes.

| Workload Modes | Arrival Rate | AVG (%) | STD (%) |
|---|---|---|---|
| Random | [0, 100] | 53.53 | 29.51 |
| Low-frequency | [20, 40] | 30.07 | 6.36 |
| High-frequency | [60, 80] | 70.32 | 5.57 |

*5.3. Experiment Results and Analysis*

The experimental results are shown in the following figures. In job arrival rate figures, which correspond to the job type distribution and job arrival rate, the line graph shows the job arrival rate, and the bar graph shows the actual number of jobs. Blue indicates compute-intensive jobs, and yellow indicates I/O-intensive jobs. The horizontal coordinate corresponds to the time (s) variation. In other figures, the average response time and QoS satisfaction, which are the focus of the SDSmp, correspond to the time (s) variation in the horizontal coordinate and the average response time or satisfaction in the vertical coordinate, and the different shapes of the line graph correspond to the performance of different methods.

In addition to the three workload modes in a shorter period, we also conducted experiments over a longer time. In addition, Table 5 shows the result in three workload modes, random, low-frequency, and high-frequency, for up to 2 h in addition to the first 40 s. The reason for removing the first 40 s was to eliminate the interference caused by the offline training phase on the real-time scheduling and formal operation. Compared with the existing methods, the proposed method can reasonably schedule the foreground jobs to Smp resources cost-consciously and improve the performance in all the different workload modes after a short learning adaptation. Specifically, it not only reduces the cost by 13.6% but also ensures load balancing, improves the quality-of-service satisfaction by 18.7%, and reduces the average response time by 34.2%.

**Table 5.** Experimental results of the different workload modes.

| Workload Modes | Metric | DQN | Random | RR | Earliest | Suitable | SensibleR |
|---|---|---|---|---|---|---|---|
| Literature | | Proposed | [28,30] | [29,33] | [10,16,31,32] | [42,45] | [44] |
| Random | Cost | 312.82 | 363.32 | 365.46 | 364.77 | 346.01 | 369.39 |
| | QoS satisfaction | 96.2% | 51.3% | 75.3% | 74.4% | 81.2% | 47.8% |
| | Balancing rate | 62.8% | 73.1% | 72.6% | 75.7% | 68.1% | 78.2% |
| | Response time | 0.203 | 0.712 | 0.426 | 0.421 | 0.275 | 1.116 |
| Low-frequency | Cost | 109.30 | 123.32 | 122.32 | 128.57 | 118.56 | 121.74 |
| | QoS satisfaction | 99.9% | 99.5% | 99.9% | 99.9% | 99.9% | 98.4% |
| | Balancing rate | 26.8% | 29.8% | 27.7% | 29.4% | 28.6% | 33.7% |
| | Response time | 0.115 | 0.237 | 0.163 | 0.158 | 0.057 | 0.254 |
| High-frequency | Cost | 556.52 | 893.13 | 895.25 | 871.77 | 817.08 | 893.14 |
| | QoS satisfaction | 93.7% | 11.4% | 12.6% | 13.8% | 70.3% | 12.2% |
| | Balancing rate | 73.2% | 98.4% | 91.7% | 97.4% | 76.8% | 98.1% |
| | Response time | 0.357 | 11.637 | 10.362 | 3.527 | 0.658 | 11.246 |

### 5.3.1. Random Workload Mode

First, a random workload model with significant variations was used to test the model's performance.

As shown in Figure 5, the job arrival rate for the random workload mode was randomly generated in the range of [0, 100]% with a mean of 266.65 (requests/s) and a standard deviation of 147.56 (requests/s). The job type was refreshed every five seconds, and the job and workload types were randomly generated.

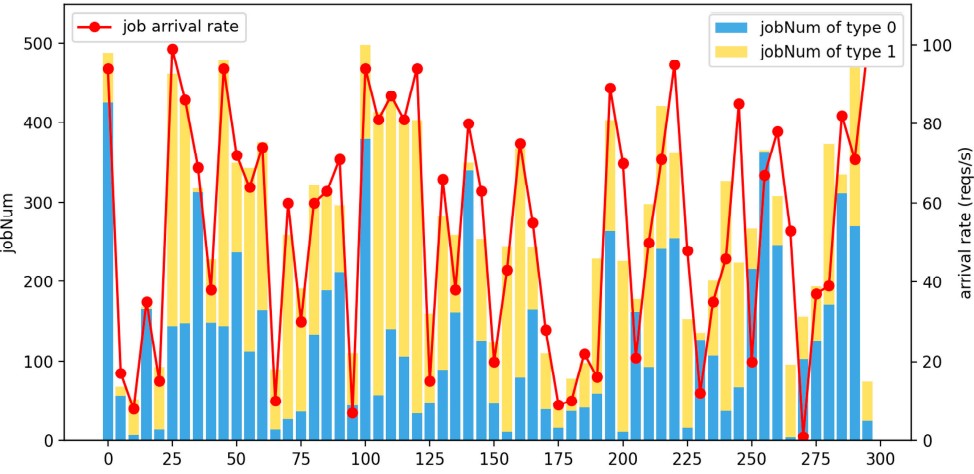

**Figure 5.** Job arrival rate in random workload mode.

As shown in Figures 6 and 7, all load queues are empty for initialization, and the first five seconds are flooded with jobs, all methods perform poorly but function, usually taking 5 to 20 s, and the requested job arrival rate is low at 8% to 35%/s; none of the methods can make up the difference and perform well. Between 25 and 125 s, due to a sudden increase in jobs and being kept in a high-frequency state, the waiting queues are overloaded and all methods are affected by the blockage. From 125 s to the end of 300 s, jobs are not continuously entered at a high frequency, the job queue is no longer severely blocked, and the Smp scheduling is orderly, with the best results for the suitable and DQN methods.

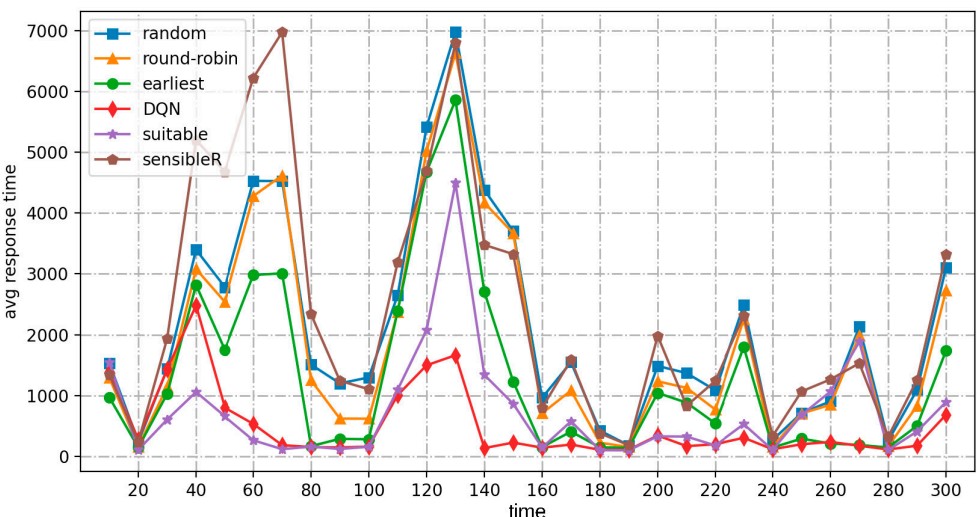

**Figure 6.** Average response time in random workload mode.

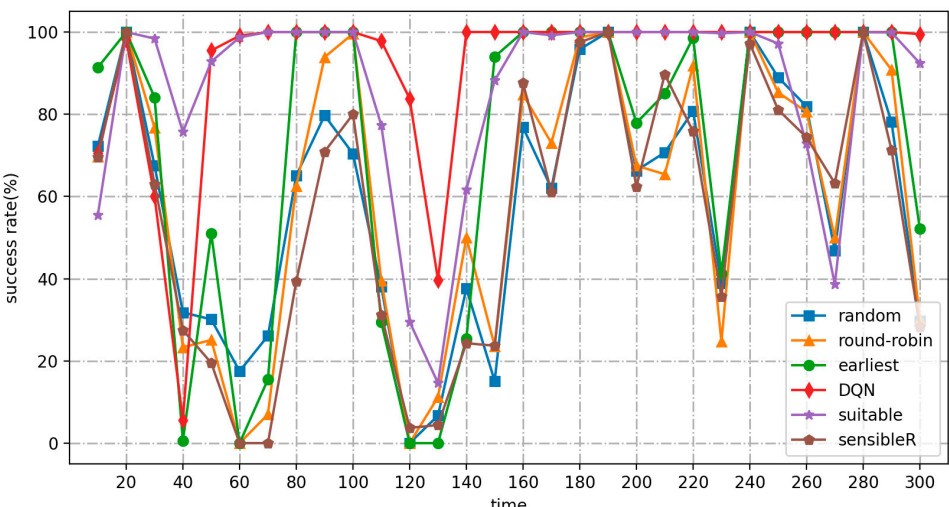

**Figure 7.** QoS satisfaction in random workload mode.

Overall, the proposed method is in an active training phase before 50 s and cannot pull apart. Around 50 s, it can be seen that the DQN method gradually completes its training, adapts to this workload pattern, and pulls away from the other methods. After that, during the high- and low-frequency variations the proposed model performs best and is better than the suitable method.

As shown in Table 5 for the random load pattern, DQN achieves the lowest cost, an average response time and load balancing rate, and the highest QoS satisfaction. Suitable performs second best, while the random and sensitive methods perform relatively poorly.

5.3.2. Low-Frequency Workload Mode

A low-frequency workload mode was set to test the algorithm's performance in the Smp's most common low-frequency idle usage scenarios.

As shown in Figure 8, the job arrival rate for the low-frequency workload model was randomly generated in the range of [20, 40] with a mean of 30.07% and a standard deviation of 6.36%.

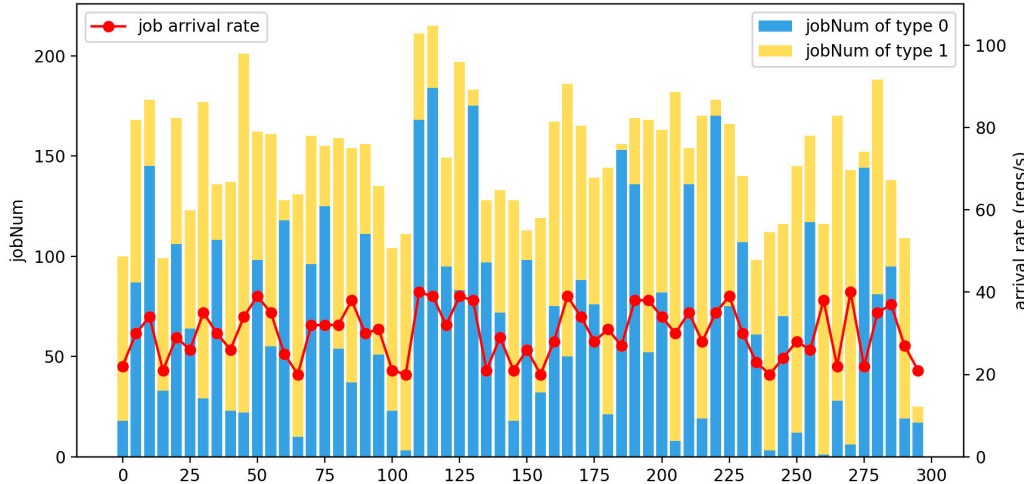

**Figure 8.** Job arrival rate in low-frequency workload mode.

Combined with the low-frequency load pattern shown in Table 5, several algorithms perform well in the low-frequency state, with generally low average response times, high QoS satisfaction, and low load balancing. The suitable method achieves the lowest average response time and the best performance; DQN has the lowest cost. As shown in Figure 9, after the initial 40 s training period, the DQN method gradually exceeds the average

response time of all methods except the suitable method. Then, it runs smoothly close to the best-performing suitable method. As shown in Figure 10, the QoS satisfaction of the six methods in the low-frequency mode fluctuates only slightly for the sensible method, and the overall performance is good.

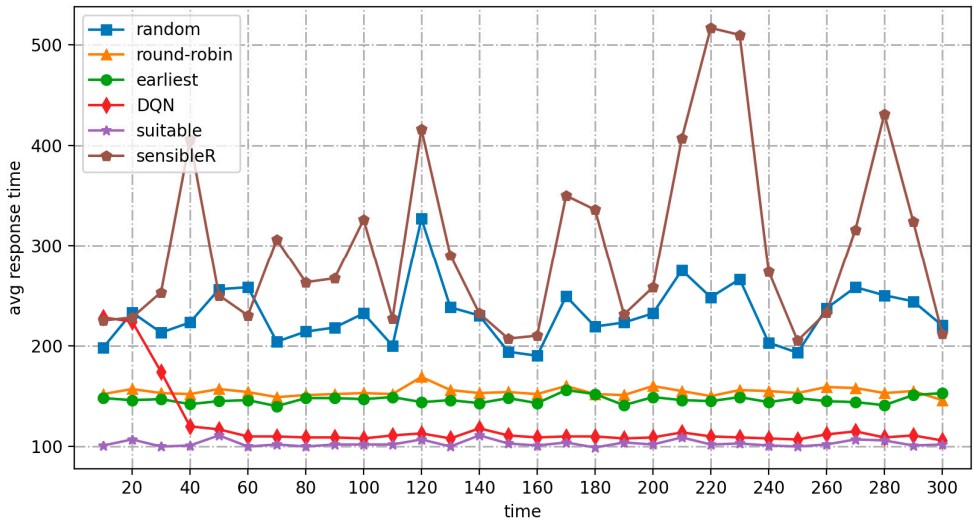

**Figure 9.** Average response time in low-frequency workload mode.

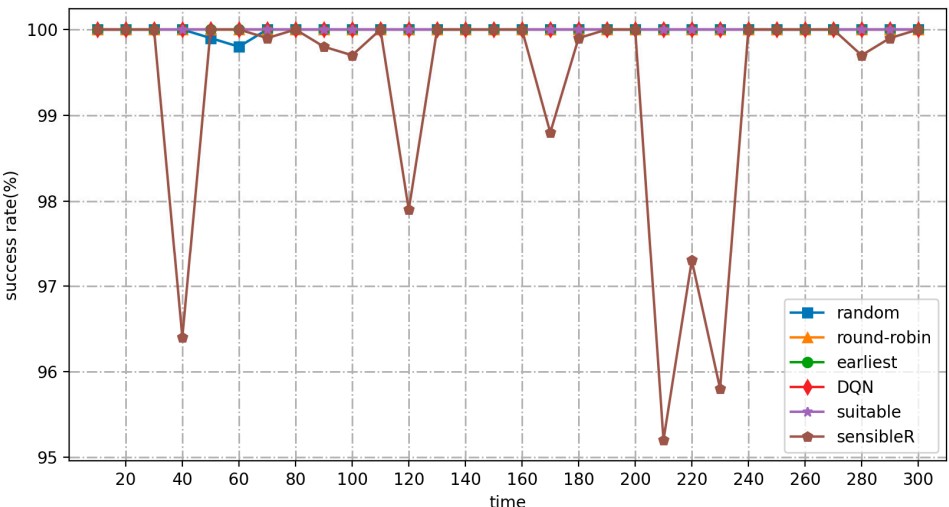

**Figure 10.** QoS satisfaction in low-frequency workload mode.

### 5.3.3. High-Frequency Workload Mode

A high-frequency workload mode was set to test the algorithm's performance in demanding usage scenarios in the Smp, such as explosive user usage and consistently high frequency.

As shown in Figure 11, the job arrival rate for the high-frequency workload model was randomly generated in the range of [60, 80] with a mean of 70.32% and a standard deviation of 5.57%.

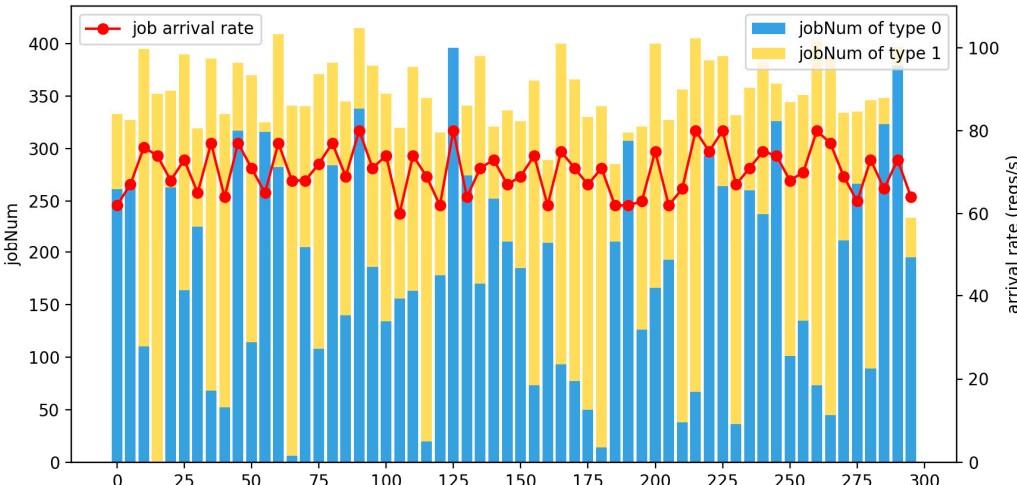

**Figure 11.** Job arrival rate in high-frequency workload mode.

As shown in Table 5 for the high-frequency load pattern, the extremely high frequency state of the settings brings the environment close to collapse, and most algorithms have difficulty adapting to this ultra-high-intensity pattern. All methods other than DQN and suitable fail to work correctly, with low load balancing rate and low QoS satisfaction performance. In contrast, DQN and suitable work normally, and suitable suffers from degraded performance due to high frequencies. DQN has the highest QoS satisfaction, lowest average response time, lowest load balancing rate, lowest cost, and superior performance in uncertain or extreme environments.

As shown in Figures 12 and 13, the average response times of random, round-robin, and earliest continue to increase until the system fails. Their QoS satisfiers are all close to 0 after 80 s. While suitable stays up and running in high-frequency workload mode, the average response time is almost always below 2500 ms. The QoS satisfiers are almost always above 40%, but fluctuate widely. The DQN algorithm, on the other hand, always performs at a very high level, with QoS satisfaction ranging from 70% to 95% for the first 30 s, but close to 100% for the rest of the time, and the training time to adapt to the new workload mode is reduced to 20 s.

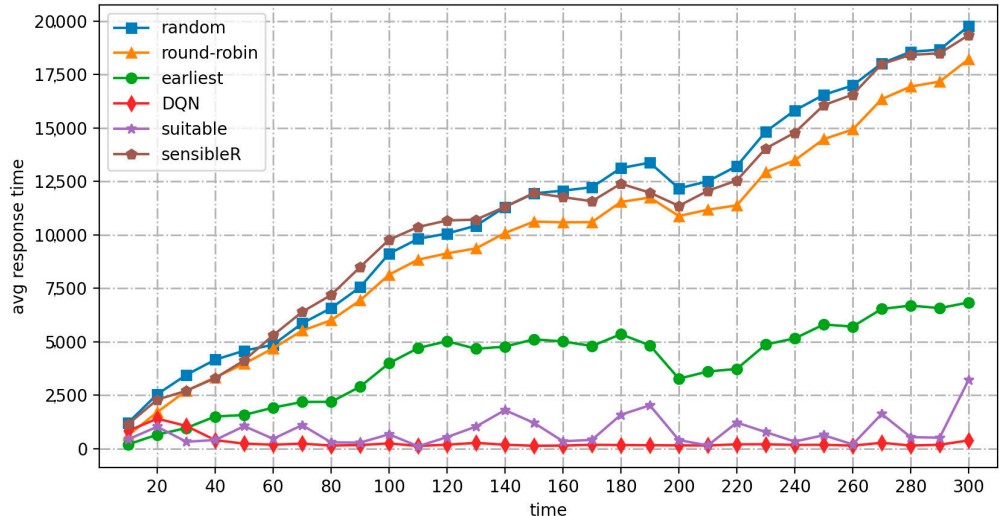

**Figure 12.** Average response time in high-frequency workload mode.

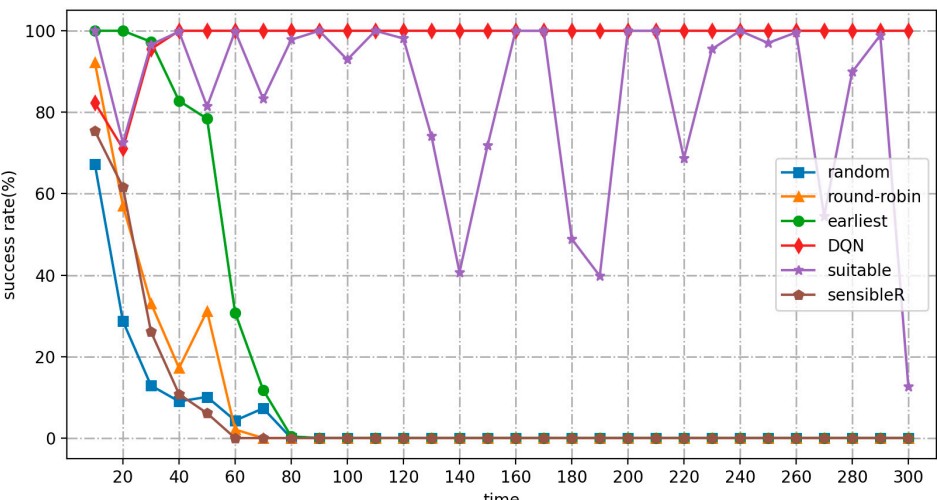

**Figure 13.** QoS satisfaction in high-frequency workload mode.

### 5.3.4. Experimental Analysis

In the experimental results of the above three workload modes, by comparing the proposed method with five existing real-time methods, it can be seen that the proposed Smp resource scheduling algorithm is suitable for various scenarios and outperforms other algorithms, and the following conclusions can be summarized:

- As the number or frequency of input jobs increases, the average response time of the DRL-based algorithm for real-time cost optimization of the MPC-SDSmp increases. Comparing the low-frequency and high-frequency workload modes, the proposed algorithm shows a more significant advantage in the high-frequency workload mode, especially when it is already obvious that the other methods do not work correctly. Suitable and the proposed algorithm still meet the availability. The proposed method is the only one among the six methods that maintains a high performance and stability, has a low average response time, the lowest load balancing rate, the lowest cost, and the highest QoS satisfaction.

- Compared to the random high-frequency workload model, the proposed algorithm is based on training experience. It has good robustness after training, making it easier to handle an unknown number of job types and more suitable for real-time environments. By encapsulating the structure, the software definition also removes the Smp from the application and infrastructure layers, improving security.

- The complete training phase from 0 s to 40 s and the subsequent execution phase during the experiment are shown in Figures 5–13 instead of showing the real-time scheduling separately from the offline pretraining. It is important to emphasize that the focus is on real-time scheduling, cost, and QoS optimization of Smp resources, not offline training. Because the offline training is done locally and does not occupy the cloud Smp resources, the consumption of Smp resources is almost 0. After the training is completed, the scheduling can be done directly. In addition to the training being completed offline, as shown in Figure 8 in low-frequency time (late at night), the Smp business and service switching process provides optional smooth online senseless deployment. The advantage of online deployment is that redeployment of the new Smp service does not interrupt the original service. There is no need to shut down the system to retrain. Only the new service needs to be online after offline training. The training copy can be senselessly switched during the regular operation of the original service, which has better scalability and fault tolerance with minimal cost difference. Therefore, the method shows good stability and robustness in the variable SDSmp environment and itself has a specific resistance to attacks and disaster recovery capabilities, making it more applicable.

- The load balancing rate metric visualizes the degree of resource utilization and overhead during the actual operation of the different schemes. As shown in Table 5, both this and the suitable method achieve advantages over existing real-time methods in terms of long-term operational performance under the three workload modes. The low-frequency simulation of the quiescent environment is performed smoothly by all methods, and the load balancing rates are similar. In the high-frequency and random load modes, because both the present method and suitable have the characteristic of learning, in the continuous operation process the other methods fall far behind. In the high-frequency load mode, the other methods enter the performance bottleneck and cannot operate normally. However, the current and suitable methods still work, proving they are still available in large-scale, high-load operation scenarios. The advantages of this method in terms of cost and load balancing are apparent.

## 6. Conclusions and Future Works

We address the security challenges and essential protection requirements of critical information infrastructure to solve the difficulty of applying existing cost-optimized solutions and performance degradation in SDSec and IoT scenarios. To address the mismatch problem between security protection means and business scenarios, we propose the SDSmp automatic control framework for fragmented security requirements and security scenarios to reduce deployment costs by improving the reuse rate of security infrastructure resources. We use MPC to protect user privacy from leakage. We propose an MPC-SDSmp cost optimization architecture to achieve real-time automated control and reduce costs. The DRL-based algorithm for real-time cost optimization of the MPC-SDSmp is proposed to further reduce the operation cost.

Furthermore, it is compared with the current mainstream real-time methods. The experimental results under three load modes show that the proposed method not only reduces the cost by 13.6% but also improves the quality-of-service satisfaction by 18.7%, reduces the average response time by 34.2%, has load balancing, and has good robustness more suitable for the real-time environment.

As future works, to enhance the effectiveness of the proposed DRL algorithm in optimizing Smp resource scheduling, our plans include applying it in a more diverse and fragmented highly dynamic real-time security scenario. We will also train our agents to address complex real-world problems, such as partial security protection failure, foreground job pre- and post-related issues, and cloud-based automatic configuration, with the aim of reducing costs even further. This will provide a more comprehensive and efficient solution for cost optimization and security enhancement in IoT and SDSec.

**Author Contributions:** Conceptualization, methodology, data collection, data analysis, and writing: Y.Q.; writing, review, editing, Y.L. All authors have read and agreed to the published version of the manuscript.

**Funding:** This work was supported in part by the State Grid Jiangxi Information & Telecommunication Company Project "Research on de-boundary security protection technology based on zero trust framework" under Grant 52183520007V.

**Data Availability Statement:** Data are available upon request.

**Conflicts of Interest:** The authors declare no conflict of interest.

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
