# Peer review of "Real-Time Cost Optimization Approach Based on Deep Reinforcement Learning in Software-Defined Security Middle Platform"

_information, doi:10.3390/info14040209_

Round 1

Reviewer 1 Report

The paper is interesting 

The abstract is written very well 

The Introduction need more discussion about the limitations 

The literature is fine, you can add the following study to the related works 

- Classification of cyber security threats on mobile devices and applications 

The methodology is clear

The results and comparison are clear

The Conclusion need rewrite 

Overall the paper has very contribution  

Author Response

Reviewer 1 Comments

Responses

(1) The paper is interesting.

The abstract is written very well.

The Introduction need more discussion about the limitations.

(1) We greatly appreciate your recognition. And we have added more discussion about the limitations in Section 1. Introduction as follows:

Line 111: Although this article makes commendable contributions, it is not without limitations. One such limitation is that, like many other deep learning applications in various fields, the training process of the proposed method is conducted offline. While offline training can help minimize costs and prevent the consumption of valuable security middle platform (Smp) resources, it also presents a limitation. However, once the model is trained, it can be deployed in real-time during normal operation without requiring additional offline training.

(2) The literature is fine, you can add the following study to the related works:

-Classification of cyber security threats on mobile devices and applications.

(2) We appreciate your suggestion and have included additional relevant references, including reference [45]:

[45] Almaiah, M. A.; Al-Zahrani, A.; Almomani, O.; Alhwaitat, A. K. Classification of cyber security threats on mobile devices and applications. In Artificial Intelligence and Blockchain for Future Cybersecurity Applications, Cham: Springer International Publishing 2021; pp. 107-123.

We have also standardized the reference style throughout the article and supplemented Section 1. Introduction and Section 2. Related work with additional research from other sources. This has not only increased the number of references but also enhanced the overall quality of the article. Thank you again for your suggestion.

(3) The methodology is clear.

The results and comparison are clear.

The Conclusion need rewrite.

Overall the paper has very contribution.

(3) We sincerely appreciate your positive comment. In response to your suggestion, we have enhanced Section 6. Conclusions and future works and included some additional future works:

Line 791: As future works, to enhance the effectiveness of the proposed deep reinforcement learning (DRL) algorithm in optimizing Smp resource scheduling, our plans include applying it in a more diverse and fragmented high dynamic real-time security scenario. We will also train our agents to address complex real-world problems, such as partial security protection failure, foreground job pre and post-related issues, and cloud-based automatic configuration, with the aim of reducing costs even further. This will provide a more comprehensive and efficient solution for cost optimization and security enhancement in Internet of Things (IoT) and software-defined security (SDSec).

Once again, thank you for your support and we wish you all the best.

Reviewer 2 Report

The authors have to address all of the below concerns carefully.

-          Paper Title: It contains some acronyms (such as MPC) that makes it unclear. We prefer to replace them with words. Also, the search title requires improvement because it is not clear.

-          Keywords: We suggest that the authors should replace keywords such as “deep reinforcement learning” and “security middle platform” because these keywords are already found in the article title. It is better that they replace them with other keywords to increase the reach of the article.

-          Acronyms must define before using in the first appearance such as MPC, NFV, API, DRL  … etc.

-          Abstract: The purpose/problem at the beginning of the abstract requires improvement to be comprehensive.

-          What is the purpose of writing the sentence “Virtualization reduces the need to allocate physical resources for high-traffic loads. Line128”?

-          Related Work Section: This section requires rewriting. It is written narratively, not critically. This section requires improvement.

-          The title "A DRL-based algorithm for real-time cost optimization of MPC-SDSmp" should be divided into sub-headings to be clearer to readers and more structured.

-          Why is the title "Performance Evaluation" repeated?

-          What is fairness in proving results and Comparative Analysis? The authors should compare their results with those of the existing research (included in this paper).

-          English Writing: This paper requires minor proofreading. There are some of grammatical, spelling and typos problems. The authors should thoroughly scrutinize the paper. Without professional, accurate and clear English, readers cannot understand the research.

- References list: References should follow the MDPI-Information style. The number of references is insufficient for this study. Some search names in the reference list begin an uppercase letter for each word (such as [2] ... etc.) and others use only an uppercase letter in the first word (such as [1] … etc.), author should standardize style. Some references do not contain enough information such as References [18] … etc. Journal names should be italicized. The references list requires extensive scrutiny by the authors.

Author Response

Reviewer 2 Comments

Responses

(1) Paper Title: It contains some acronyms (such as MPC) that makes it unclear. We prefer to replace them with words. Also, the search title requires improvement because it is not clear.

(1) Thank you for your valuable comments. In order to improve the clarity of the title, we have made changes to it by removing the use of acronyms. The revised title is “A Deep Reinforcement Learning Based Algorithm for Real-Time Cost Optimization of Software-Defined Security Middle Platform”.

The purpose of these changes is to enhance the clarity of the main objectives and methods of our work for the readers, with the remaining details expounded upon in the body of the paper. Moreover, we have conducted revisions to all search titles as follow:

1. Introduction;

2. Related work;

3. Our scheduling model;

3.1. Foreground job characteristics;

3.2. Security middle platform resources;

3.3. Job scheduling mechanism;

4. Our DRL-based scheduling;

5. Evaluation;

5.1. Experimental framework;

5.2. Experiment results;

5.2.1. Random workload mode;

5.2.2. Low-frequency workload mode;

5.2.3. High-frequency workload mode;

5.2.4. Experimental analysis;

6. Conclusions and future works.

(2) Keywords: We suggest that the authors should replace keywords such as “deep reinforcement learning” and “security middle platform” because these keywords are already found in the article title. It is better that they replace them with other keywords to increase the reach of the article.

(2) Thanks for the comment. To avoid repetition and broaden the coverage of the article, we have revised the keywords as follows:

software-defined security; deep reinforcement learning; cost optimization; internet of things; privacy protection.

(3) Acronyms must define before using in the first appearance such as MPC, NFV, API, DRL… etc.

(3) We sincerely appreciate your valuable suggestion. To eliminate any potential confusion, we have meticulously reviewed all acronyms and provided definitions upon their first usage:

Deep reinforcement learning (DRL);

Security middle platform (Smp);

Software-defined security middle platform (SDSmp);

Software-defined security (SDSec);

Software-defined networks (SDN);

Internet of things (IoT);

Network functions virtualization (NFV);

Multi-Party Computation (MPC);

Application Programming Interface (API).

(4) Abstract: The purpose/problem at the beginning of the abstract requires improvement to be comprehensive.

(4) We sincerely appreciate your valuable suggestion. To enhance the clarity and coherence of our article, we have carefully rephrased the abstract to better articulate the purpose and research questions of our study. Please find the revised abstract below:

Line 8: In today's business environment, reducing costs is crucial due to the variety of Internet of Things (IoT) devices and security infrastructure. However, applying security measures to complex business scenarios can lead to performance degradation, making it a challenging task. To overcome this problem, we propose a novel algorithm based on deep reinforcement learning (DRL) for optimizing cost in Multi-Party Computation software-defined security middle platforms (MPC-SDSmp) in real-time. To accomplish this, we first integrate fragmented security requirements and infrastructure into the MPC-SDSmp cloud model with privacy protection capabilities to reduce deployment costs. By leveraging the power of DRL and cloud computing technology, we enhance the real-time matching and dynamic adaptation capabilities of the security middle platform (Smp). This enables us to generate a real-time scheduling strategy for Smp resources that meet low-cost goals to reduce operating costs. Our experimental results demonstrate that the proposed method not only reduces the costs by 13.6%, but also ensures load balancing, improves the quality of service satisfaction by 18.7%, reduces the average response time by 34.2%. Moreover, our solution is highly robust and better suited for real-time environments than existing methods.

(5) What is the purpose of writing the sentence “Virtualization reduces the need to allocate physical resources for high-traffic loads. Line128”?

(5) We sincerely appreciate your attention to detail and valuable suggestion. In response to your comments, we have made necessary revisions to rectify any inaccuracies in the article. Please find the updated version of paragraph 128 below:

Line 150: Virtualizing security infrastructure reduces physical deployment, making it particularly suitable for high-traffic scenarios. Virtualized security solutions are cost-effective because they optimize network resources, and security resources are available on demand. Network capital costs are reduced by optimizing resources and intermediate devices in a virtual environment, significantly reducing the resources required for security.

(6) Related Work Section: This section requires rewriting. It is written narratively, not critically. This section requires improvement.

(6) Thank you for your valuable suggestion. We have taken it into consideration and made necessary revisions to Section 2. Related work. Additional references have been incorporated to underscore the innovation and importance of our work, while also critically evaluating relevant studies. For example:

Line 134: However, existing works have focused on optimizing the use of partial network resources for software-defined security (SDSec) and software-defined networks (SDN) by adopting various resource scheduling algorithms from different perspectives, without considering the fragmented security requirements and scenarios that are prevalent in the entire security field. As a result, they still need to address the problem of a mismatch between security measures and business scenarios. We take a holistic approach to network security and propose the design of the MPC-SDSmp architecture to target and solve these issues specifically.

Line 154: They have successfully optimized deployment costs, but to fully address security concerns, it is necessary to consider cost optimization of security facilities during operation and user privacy leaks.

Line 171: Existing researches have highlighted the fragmented security needs and the risk of privacy leaks in security scenarios. Still, they have yet to address these issues from the perspective of cost optimization, security capability, and user privacy protection. Traditional methods have also failed to prevent privacy leaks fundamentally. We propose MPC-SDSmp to reduce deployment costs and protect user privacy.

Line 190: To improve the efficiency of security resource scheduling in the security service chain, it is essential to effectively balance the load of various virtual security devices. This presents a challenging task; the proposed algorithm based on DRL and MPC-SDSmp aims to address this issue. The algorithm optimizes resource scheduling to improve the utilization of Smp resources and reduce operating costs while protecting user privacy and ensuring security capabilities. Overall, the algorithm provides a comprehensive solution for cost optimization and security enhancement in IoT and SDSec.

(7) The title "A DRL-based algorithm for real-time cost optimization of MPC-SDSmp" should be divided into sub-headings to be clearer to readers and more structured.

(7) Many thanks for the suggestion. We have incorporated your suggestions and made improvements to our article. We have modified the subheading of Section 4 to “Our DRL-based scheduling”. Specifically, in Section 3. Our scheduling model, we have proposed an SDSmp automatic control architecture for SDSec and IoT, and have utilized an advanced DRL algorithm to optimize the scheduling of security infrastructure in Section 4. Our DRL-based scheduling. Our primary objective is to reduce both deployment and operation costs, while simultaneously ensuring robust security protection and safeguarding user privacy. Additionally, we have reorganized all subheadings to enhance the clarity and coherence of the article. Once again, we appreciate your valuable suggestion.

(8) Why is the title "Performance Evaluation" repeated?

(8) Thank you for carefully reading our article. Based on your suggestion, we have made adjustments to all of the subheadings in order to improve the clarity and organization of the content. Please find the revised subheadings listed below:

1. Introduction;

2. Related work;

3. Our scheduling model;

3.1. Foreground job characteristics;

3.2. Security middle platform resources;

3.3. Job scheduling mechanism;

4. Our DRL-based scheduling;

5. Evaluation;

5.1. Experimental framework;

5.2. Experiment results;

5.2.1. Random workload mode;

5.2.2. Low-frequency workload mode;

5.2.3. High-frequency workload mode;

5.2.4. Experimental analysis;

6. Conclusions and future works.

(9) What is fairness in proving results and Comparative Analysis? The authors should compare their results with those of the existing research (included in this paper).

(9) Thank you for your valuable suggestion. We have modified the subheading of Section 5.2.4. to “Experimental analysis”. In Section 5. Evaluation, we compare our proposed method with five of the most widely used real-time scheduling algorithms: random scheduling method, round-robin scheduling method, earliest scheduling method, suitable scheduling method, and sensible scheduling method. We conduct this comparison in three different workload modes: random workload mode (in Section 5.2.1.), low-frequency workload mode (in Section 5.2.2.), and high-frequency workload mode (in Section 5.2.3.). We provide a clear demonstration of the experimental process and results. In Section 5.2.4. Experimental analysis, we perform a more detailed analysis of the experimental results and prove the superiority of our proposed method by comparing its performance under different workload modes and scheduling algorithms.

(10) English Writing: This paper requires minor proofreading. There are some of grammatical, spelling and typos problems. The authors should thoroughly scrutinize the paper. Without professional, accurate and clear English, readers cannot understand the research.

(10) Many thanks for the comments, we have proofread the revised manuscript for further improving the presentation.

(11) References list: References should follow the MDPI-Information style. The number of references is insufficient for this study. Some search names in the reference list begin an uppercase letter for each word (such as [2] ... etc.) and others use only an uppercase letter in the first word (such as [1] … etc.), author should standardize style. Some references do not contain enough information such as References [18] … etc. Journal names should be italicized. The references list requires extensive scrutiny by the authors.

(11) Thanks a lot for the suggestion. We have taken your suggestions into consideration and made improvements to the paper. Specifically, we have followed the MDPI style guidelines to standardize the style of all references and added more references to enhance the quality of the paper. Additionally, we have included further studies in Section 1. Introduction and Section 2. Related work. Thank you once again for your suggestions and we wish you all the best.

Reviewer 3 Report

The study seems interesting. However, in order to improve the quality of manuscript, my comments are given below.

1) The authors are advised to add motivation, contribution, and the benefits of your research in the introduction section.

2) In related work please add the most recent studies in this area; ie. 2022. for example; 

Abbasi R, Mateen A, Ali Abid M, Khan S. A Step toward Next-Generation Advancements in the Internet of Things Technologies. Sensors. 2022; 22(20):8072. https://doi.org/10.3390/s22208072

3) Please add a table in section 4 and mention all mathimatical symbols along with the description.

4) Please improve the quality of the figures in the manuscript.

5) Please mention future work in the conclusion section.

***** Good luck.****

Author Response

Reviewer 3 Comments

Responses

(1) The authors are advised to add motivation, contribution, and the benefits of your research in the introduction section.

(1) Thank you for your valuable comment. To further enhance the motivation of our research and provide a clearer understanding of the problem and goals, we have improved Section 1. Introduction, for example:

Line 31: With the proliferation of Internet of things (IoT) devices, vast amounts of data are being generated, and the number and types of these devices will continue to expand in the future. As a result, traditional IoT systems may not be equipped to adequately handle the associated challenges [6].

Line 37: In other words, As the variety and quantity of IoT devices and security infrastructure continue to increase rapidly, cost reduction has become the most pressing challenge for organizations. However, the mismatch between security measures and business scenarios presents a critical issue in cost optimization.

Line 74: The current protocols, including Azure IoT [11], are based on cloud computing and may not be able to meet the quality of service requirements of IoT systems.

Moreover, we have outlined the specific contributions of this paper as follows:

Line 88: To address the cost optimization challenges associated with software-defined security (SDSec), we have made significant efforts to reduce the cost of software-defined security middle platform (SDSmp). Contributions can be summarized as follows:

Architecturally, it reduces deployment costs by optimizing the architecture and increasing the reuse of security infrastructure resources. Specifically, SDSmp proposes an automated control architecture for fragmented security requirements and security scenarios, realizes real-time scheduling and automatic control of security middle platform (Smp) resources, and makes the security infrastructure physically and geographically independent through network functions virtualization (NFV) and cloud computing technologies. Multi-Party Computation (MPC) ensures that the security application layer is data agnostic and protects user privacy from leakage, enabling the security infrastructure to achieve resource reuse by building Smp.

In terms of modeling, an SDSmp cost optimization model is established based on deep reinforcement learning (DRL) algorithms so that the intelligent scheduler in the control plane can learn how to rationally select Smp resources based on real-time experience. This reduces operational costs and achieves high quality of service satisfaction, low response time, and load balancing.

An experimental SDSmp environment is built for implementation. The proposed DRL-based algorithm for real-time cost optimization of MPC-SDSmp is compared with existing real-time job scheduling algorithms under different workload patterns. The experimental results show that the proposed method outperforms existing real-time methods regarding cost, average response time, quality of service (QoS) satisfaction, and load balancing.

(2) In related work please add the most recent studies in this area; ie. 2022. for example;

Abbasi R, Mateen A, Ali Abid M, Khan S. A Step toward Next-Generation Advancements in the Internet of Things Technologies. Sensors. 2022; 22(20):8072. https://doi.org/10.3390/s22208072

(2) We appreciate your suggestion and have included additional relevant references, including reference [6]:

[6] Amin, F.; Abbasi, R.; Mateen, A.; Ali Abid, M.; Khan, S. A step toward next-generation advancements in the internet of things technologies. Sensors 2022, 22, 8072.

We have also standardized the reference style throughout the article and supplemented Section 1. Introduction and Section 2. Related work with additional research from other sources. This has not only increased the number of references but also enhanced the overall quality of the article. Thank you again for your suggestion.

(3) Please add a table in section 4 and mention all mathimatical symbols along with the description.

(3) Thank you for your valuable suggestion. To enhance the clarity of our paper, we have added Table 2. Notations used in our DRL-based scheduling based on Table 1. This table summarizes all the symbols used in Section 4. Our DRL-based scheduling for the convenience of readers.

(4) Please improve the quality of the figures in the manuscript.

(4) Thank you for your valuable suggestion. To enhance the clarity of our manuscript, we have taken your suggestion into account and made several improvements, such as redrawing high-precision versions of Figures 1, 2, and 3, improving the quality of the figures, correcting errors, and addressing issues of dense text. We appreciate your suggestion and thank you again for your valuable contribution.

(5) Please mention future work in the conclusion section.

(5) Your suggestion is greatly appreciated. We have carefully reviewed your suggestion and have taken steps to address it. Specifically, we have included future work in Section 6. Conclusions and future works to enhance the overall quality of the paper. Thank you for your valuable suggestions and for helping us to improve our manuscript.

Line 791: As future works, to enhance the effectiveness of the proposed DRL algorithm in optimizing Smp resource scheduling, our plans include applying it in a more diverse and fragmented high dynamic real-time security scenario. We will also train our agents to address complex real-world problems, such as partial security protection failure, foreground job pre and post-related issues, and cloud-based automatic configuration, with the aim of reducing costs even further. This will provide a more comprehensive and efficient solution for cost optimization and security enhancement in IoT and SDSec.

Round 2

Reviewer 2 Report

The authors should address all of the below concerns carefully.

-          Paper Title: The search title requires improvement because it is not clear.

-          Related Work Section: This section still requires rewriting. It is written narratively, not critically.

-          The title "Our DRL-based scheduling" should be divided into sub-headings to be clearer to readers and more structured.

-          What is fairness in proving results and Comparative Analysis? The authors should compare their results with those of the existing research (included in this paper). This comment has not been responded to. List the references in the figures to make the comparison clear.

Author Response

Reviewer 2 Comments

Responses

(1) Paper Title: The search title requires improvement because it is not clear.

(1) Thank you for your valuable feedback. We have revised the title of our paper to: “Real-Time Cost Optimization Approach Based on Deep Reinforcement Learning in Software-Defined Security Middle Platform.” Our proposed approach utilizes an enhanced deep reinforcement learning algorithm to optimize costs in real-time. It is specifically designed for the software-defined security middle platform, demonstrating its potential to improve security while optimizing costs.

(2) Related Work Section: This section still requires rewriting. It is written narratively, not critically.

(2) Thank you for your valuable comments. We have rewritten Related Work based on your feedback.

Conventional approaches: In the past few decades, extensive research has been conducted on optimizing security infrastructure and IoT devices, and efforts are underway to enhance traditional methods. To address IoT heterogeneity, unify security infrastructure and IoT devices, decouple security operations and security control [27], and achieve unified management of security devices, various custom frameworks have been developed [10] [16] [27] [28] [29] [30] [31] [32]. However, these frameworks are all based on cybernetic methods, which have limited performance improvements and are unsuitable for dynamic security scenarios. Round-robin methods in [29] [33] suffer from severe delays and poor service quality, which is unacceptable for real-world security protection scenarios. Furthermore, the [30] method lacks elasticity and scalability to meet the needs of modern network security.

On the other hand, optimization algorithms based on linear programming and fixed strategies [34] [35] [36] [37], as well as metaheuristics [38] [39] [40] [41] [42] [43] [44] [45], have demonstrated their powerful capabilities in optimizing resource usage and job processing time. For instance, analyses on DDoS attacks in software-defined security (SDSec) [26] [28] [46] have achieved security protection through access control policies. However, [35] is strictly limited with limited applicability scenarios, and high-dimensional vectors learned from the source domain are unsuitable for the target domain. Likewise, [34] is strictly limited and unsuitable for highly dynamic security scenarios. While [36] constructs an anomaly detection module and a multi-level security response module to deal with various attacks, the physical infrastructure of security protection needs to be redesigned, and the control policies in the controller need to be reprogrammed, making it challenging to deploy. As for [37], security protection is implemented in a single kernel, with a narrow scope of application and easy to reach performance bottlenecks.

The heterogeneity of security infrastructure and IoT devices implies that deploying and configuring appropriate security mechanisms requires significant overhead. These methods often have strict limitations and cannot be used in different scenarios. Almost all conventional methods aim to address batch processing jobs, which are unsuitable for real-time, highly dynamic security capability services in processing transaction security middle platform (Smp) workloads, due to the huge overhead of solving optimization problems.

Although particular methods have been developed to manage computing resources and jobs autonomously and interactively based on the state of security systems, such as the Monitor-Analyze-Plan-Execute (MAPE) loop [21] [47] [48] [49] [50] [51] [52]. Although monitoring the execution of security capabilities in software-defined network infrastructure is possible [48], it has limited functionality. [49] SDSec enhances the information security of vehicular ad hoc networks in large-scale wireless environments with high dynamic topology, but its applicability is limited. This method [50] is unsuitable for rapidly changing security environments due to poor controller interaction. Therefore, their planning phase still relies on solving complex optimization problems. These problems have limited applicability and flexibility, making them unsuitable for highly dynamic and real-time security environments.

DRL-based methods: In contrast, deep reinforcement learning (DRL) methods have demonstrated high accuracy and the ability to handle complex control problems with high-dimensional state space and low-dimensional action space using deep neural networks [56]. DRL technology can effectively address complex decision-making problems [53] [54] [55], requiring only minimal training to solve various optimization problems [57]. Indeed, [25] proposes a method for optimizing quality of service in the cloud and suggests that DRL-based algorithms are effective for cloud job scheduling in scenarios with variable workloads and complex decision-making.

Furthermore, reinforcement learning algorithms have been employed in other security-related fields to optimize routing and improve throughput [58] [59] [60], enhance the accuracy of multi-class classification tasks in intrusion detection [61], defend against Distributed Denial of Service (DDoS) attacks in software-defined network (SDN) [62], and improve system load balancing [63]. In contrast to these previous works, our research aims to use state-of-the-art DRL techniques to schedule heterogeneous security infrastructure and IoT devices to reduce deployment and operational costs. This represents a new area of research in SDSec and the IoT.

Efficient resource scheduling optimization is fundamental to improving the efficiency of highly dynamic, real-time heterogeneous security infrastructure. This presents a challenging task. The proposed algorithm based on DRL and multi-party computation software-defined security middle platforms (MPC-SDSmp) aims to address this issue. The algorithm optimizes resource scheduling to improve the utilization of Smp resources and reduce operating costs while protecting user privacy and ensuring security capabilities. Overall, the algorithm provides a comprehensive solution for cost optimization and security enhancement in IoT and SDSec.”

(3) The title "Our DRL-based scheduling" should be divided into sub-headings to be clearer to readers and more structured.

(3) Many thanks for your comments. Based on your feedback, we have reorganized the subheadings in Section 4 as follows:

4. Methodology.

4.1. Basics of DRL.

4.2. Our DRL-based scheduling.

4.2.1. Action space.

4.2.2. State space.

4.2.3. Action selection and state transition.

4.2.4. Reward function.

4.3. Training phase.

(4) What is fairness in proving results and Comparative Analysis? The authors should compare their results with those of the existing research (included in this paper). This comment has not been responded to. List the references in the figures to make the comparison clear.

(4) We sincerely apologize for not fully understanding your previous feedback. In response to your suggestions, we have made significant improvements, and you can find the relevant modifications in the article or appendices.

a) Firstly, we have added a new sub-heading, 5.2 "Baseline Solutions," to mark the mainstream real-time baseline solutions. This allows the readers to clearly understand the baseline solutions and we provided all the reference sources for them.

b) Secondly, we have supplemented all the reference sources for the comparison methods in table 5.

c) Thirdly, we have listed the reference sources for the baseline solutions in all experimental results figures (6-13).

d) To further clarify any doubts about fairness, we would like to explain that we design our experiments, including the workload modes, metrics, and baseline solutions, according to [25]. The relevant experimental settings are meticulously detailed in sections 5.1, "Experimental Framework," and 5.2, "Baseline Solutions." These metrics are standard, and the workload modes are designed to simulate real-world scenarios, which is reasonable.

We have compared our method with five existing real-time baseline methods, including Random, Round-robin, Earliest, Suitable, and Sensible. All methods were compared under the same job arrival rate in three workload modes, including Random, Low-frequency, and High-frequency, as highlighted in bold in table 4. We comprehensively displayed all the details of average response time and QoS satisfaction to demonstrate the algorithm's performance.

To ensure fairness and eliminate accidental results, we conducted experiments with longer time windows. The experimental results are shown in Table 5, "Experimental results of the different workload modes." Four common metrics, including cost, QoS satisfaction, average response time, and load balancing rate, which are highlighted in bold in the text, were used to measure the algorithm's performance. The results of the three different workload modes and the long-term experimental results can be clearly seen in section 5.3, " Experiment results and analysis." In addition, we evaluate the performance of the proposed algorithm in each workload mode section. Moreover, section 5.3.4, "Experimental Analysis," offers a more comprehensive analysis of the performance of our method across different workload modes. All of these works are taken to ensure fairness as much as possible.

Once again, thank you for your contributions and assistance with my manuscript. If you have any additional suggestions or feedback on the revised manuscript, please do not hesitate to let me know. We welcome any feedback to make my manuscript even better.

Appendix of response 4:

a) We have added references in Section 5.2. Baseline solutions.

5.2. Baseline solutions

To evaluate the performance of the proposed MPC-SDSmp cost optimization architecture based on DRL (denoted as DQN), we compare it with five other standard methods: random scheduling method [28] [30], round-robin scheduling method [29] [33], earliest scheduling method [10] [16] [31] [32], suitable scheduling method [42] [45], and sensible scheduling method [44].

Among the standard cybernetic scheduling algorithms, the random scheduling method [28] [30] is straightforward and chooses a random VM instance for each job. The round-robin scheduling approach focuses primarily on scheduling jobs to VM instances. As a result, VM instances are selected in a round-robin [29] [33] order to execute incoming jobs. The earliest scheduling method [10] [16] [31] [32] is a first-come, first-served policy in which newly arriving jobs are scheduled on the earliest available VM instance.

The suitable scheduling method [42] [45] is a greedy algorithm that tries to make the best choice. Unlike the earliest scheduling methods, the suitable scheduling method considers two factors, the time factor and whether the type of the selected VM instance matches the type of the newly arrived job. It always reduces execution time by finding the local optimum, not the overall optimum, and assigning the job to the correct type of VM instance. This means the suitable scheduling method assigns newly arrived jobs to the first busy VM instance of the correct type.

The sensible scheduling method [44] is an adaptive heuristic algorithm that uses a random routing policy based on the expected QoS, i.e., the average job response time. Jobs are assigned to VM instances with a higher probability of a lower average response time. The sensible scheduling method requires a continuous observation time D and a discount factor a. The experimental settings are D=5s, a=0.7, and D=0.2s, a=0.7.”

b) We have added references in Table 5.

Table 5. Experimental results of the different workload modes.

Workload modes

Metric

DQN

Random

RR

Earliest

Suitable

SensibleR

Literature

Proposed

[28][30]

[29[33]

[10][16][31][32]

[42][45]

[44]

Random

Cost

312.82

363.32

365.46

364.77

346.01

369.39

QoS satisfaction

96.2%

51.3%

75.3%

74.4%

81.2%

47.8%

Balancing rate

62.8%

73.1%

72.6%

75.7%

68.1%

78.2%

Response time

0.203

0.712

0.426

0.421

0.275

1.116

low-frequency

Cost

109.30

123.32

122.32

128.57

118.56

121.74

QoS satisfaction

99.9%

99.5%

99.9%

99.9%

99.9%

98.4%

Balancing rate

26.8%

29.8%

27.7%

29.4%

28.6%

33.7%

Response time

0.115

0.237

0.163

0.158

0.057

0.254

High-frequency

Cost

556.52

893.13

895.25

871.77

817.08

893.14

QoS satisfaction

93.7%

11.4%

12.6%

13.8%

70.3%

12.2%

Balancing rate

73.2%

98.4%

91.7%

97.4%

76.8%

98.1%

Response time

0.357

11.637

10.362

3.527

0.658

11.246

c) In Figures 6-13, we mark the references for the baseline solutions.

Figure 5. Job arrival rate in random workload mode.

Figure 6. Average response time in random workload mode.

Figure 7. QoS satisfaction in random workload mode.

Figure 8. Job arrival rate in low-frequency workload mode.

Figure 9. Average response time in low-frequency workload mode.

Figure 10. QoS satisfaction in low-frequency workload mode.

Figure 11. Job arrival rate in high-frequency workload mode.

Figure 12. Average response time in high-frequency workload mode.

Figure 13. QoS satisfaction in high-frequency workload mode.

d) We have designed our experiments, including the workload modes, metrics, and baseline solutions, according to [25]. The relevant experimental settings are meticulously detailed in sections 5.1, "Experimental Framework," and 5.2, "Baseline Solutions."

“We use four different metrics to evaluate the performance of each method [25]. The first metric is QoS satisfaction, which measures how many jobs are completed. This scheduling satisfies the QoS requirements, and the scheduling is successful if and only if the response time of a job is less than the predefined QoS requirements. The second metric is average response time, which measures the average time it takes to process each job. The third metric is cost, which measures operating all Smp resources. The fourth metric is the load balancing rate, which measures Smp resources utilization. Generally, the lower the load balancing rate, the better the scheduling method is. In other words, to handle jobs of the same intensity, an efficient scheduling method will use fewer resources in the scheduling process and ultimately have a lower load-balancing rate.

In addition, we set up three different workload patterns [25].”

Table 4. Generation of load modes.

Workload modes

Arrival Rate

AVG(%)

STD(%)

Random

[0,100]

53.53

29.51

Low-frequency

[20,40]

30.07

6.36

High-frequency

[60,80]

70.32

5.57

Reference:

[10] Al-Ayyoub, M.; Jararweh, Y.; Benkhelifa, E.; Vouk, M.; Rindos, A. Sdsecurity: A software defined security experimental framework. In Proceedings of the 2015 IEEE International Conference on Communication Workshop (ICCW), June 2015; pp. 1871-1876.

[16] Kim, Y.; Nam, J.; Park, T.; Scott-Hayward, S.; Shin, S. SODA: A software-defined security framework for IoT environments. Computer Networks. 2019, 163, 106889.

[25] Wei, Y.; Pan, L.; Liu, S.; Wu, L.; Meng, X. DRL-scheduling: An intelligent QoS-aware job scheduling framework for applications in clouds. IEEE Access. 2018, 6, 55112-55125.

[28] El Moussaid, N.; Toumanari, A.; El Azhari, M. Security analysis as software-defined security for SDN environment. In Proceedings of the 2017 Fourth International Conference on Software Defined Systems (SDS), May 2017; pp. 87-92.

[29] Liang, X.; Qiu, X. A software defined security architecture for SDN-based 5G network. In Proceedings of the 2016 IEEE International Conference on Network Infrastructure and Digital Content (IC-NIDC), September 2016; pp. 17-21.

[30] Liyanage, M.; Ahmed, I.; Ylianttila, M.; Santos, J. L.; Kantola, R.; Perez, O. L.; Jimenez, C. Security for future software defined mobile networks. In Proceedings of the 2015 9th International Conference on Next Generation Mobile Applications, Services and Technologies, September 2015; pp. 256-264.

[31] Luo, S.; Salem, M. B. Orchestration of software-defined security services. In Proceedings of the 2016 IEEE International Conference on Communications Workshops (ICC), May 2016; pp. 436-441.

[32] Farahmandian, S.; Hoang, D. B. SDS 2: A novel software-defined security service for protecting cloud computing infrastructure. In Proceedings of the 2017 IEEE 16th International Symposium on Network Computing and Applications (NCA), October 2017; pp. 1-8.
